# Quantum EigenGame for excited state calculation

David Quiroga,* Jason Han, Anastasios Kyrillidis
Computer Science, Rice University and Ken Kennedy Institute
`daq3@rice.edu, jh146@rice.edu, anastasios@rice.edu`

Computing the excited states of a given Hamiltonian is computationally hard for large systems, but methods that do so using quantum computers scale tractably. This problem is equivalent to the PCA problem where we are interested in decomposing a matrix into a collection of principal components. Classically, PCA is a well-studied problem setting, for which both centralized and distributed approaches have been developed. On the distributed side, one recent approach is that of EigenGame, a game-theoretic approach to finding eigenvectors where each eigenvector reaches a Nash equilibrium either sequentially or in parallel. With this work, we extend the EigenGame algorithm for both a $0^{\text{th}}$-order approach and for quantum computers, and harness the framework that quantum computing provides in computing excited states. Results show that using the Quantum EigenGame allows us to converge to excited states of a given Hamiltonian without the need of a deflation step. We also develop theory on error accumulation for finite-differences and parameterized approaches.

## 1. Introduction

Quantum computing offers an alternative approach to solving complex computational tasks, potentially reducing the time and space complexity compared to classical methods. Quantum algorithms –like Quantum Phase Estimation [1], the Deutsch-Jozsa algorithm [2], and Grover's algorithm [3]– demonstrate superior performance in ideal, noiseless conditions. However, in the Noisy Intermediate-Scale Quantum (NISQ) era [4], noise remains a significant challenge, influencing the stability and reliability of quantum computations [5–8].

Performing optimization tasks under noisy settings is a common scenario in the algorithmic literature. In optimization and machine learning, errors that propagate throughout iterations critically influence performance metrics and outcomes [9–12]. Understanding and mitigating error propagation is crucial for enhancing the practical utility of algorithms in real-world applications.

Particularly relevant to the present work, consider the case of derivative-free optimization (DFO) [13–18]: DFO is employed effectively in scenarios where traditional gradient-based methods falter [16]. However, the efficiency of DFO methods often lags, particularly for high-dimensional problems, due to their reliance on sampling routines that may require many function evaluations to approximate gradients [15]. Further, DFO may struggle with precision near minima [17]. Overall, DFO demands more resources, posing a significant limitation in large-scale scenarios.

**This paper's focus: VQE.** Within this interplay between algorithms and error in calculations, we focus on a specific type of variational quantum algorithm [19], the Variational Quantum Eigensolver (VQE) [20]. VQE is the quantum analog of PCA that approximates the minimum/maximum eigenvalue/eigenvector pair of a given matrix (Hamiltonian), and can provide exponential time and space reduction over classical methods [21]. This problem is crucial for tasks like finding the ground state energies of molecules, which is pivotal in fields ranging from quantum chemistry to materials science [21–23]. Yet, VQE constitutes a framework with multiple moving parts: classical optimization procedures, quantum ansatzes, quantum observables, and hyperparameter tuning, all requiring careful setup before being applied to specific domains.

---

*Corresponding author

Second Conference on Parsimony and Learning (CPAL 2025).

While VQE is meant for calculating the ground state energy of a Hamiltonian, it is also of interest to higher energy states, also known as "excited states" [24–26]. Computing excited states provides insight into the properties of quantum systems. In quantum chemistry, determining excited states helps to understand chemical reaction mechanisms, including the absorption spectra and photochemistry of molecules [27]. In materials science, excited states can reveal information about electronic properties such as conductivity and magnetism, which are vital to designing new materials with targeted functionalities [28]. Calculating these excited states, as opposed to only finding the ground state, provides broader information on molecules of interest, and poses a greater technical challenge due to the requirement of maintaining orthogonality among calculated states.

**Motivation and our contributions.** This work builds upon a classical collaborative computation model, the EigenGame, presented in [29], to compute excited states in a noisy quantum environment. Unlike existing approaches that rely on deflation to calculate successive states [30–35], our method allows such a computation via a regularized objective inspired by game-theoretic ideas. Our primary contributions are:

- *Algorithmic Framework*: We propose a VQE meta-algorithm that changes the computational dynamics to find excited states. Using a modified quantum objective, our approach avoids deflation steps, diverging from traditional deflation-based computation models.
- *Theoretical Component*: We provide a theory that validates the convergence properties of our algorithm. By formalizing the interaction between error calculations –due to noisy gradients and DFO routines– and convergence rates, we establish a theory for VQE targeting excited states.
- *Empirical Validation*: The current implementation shows that our approach leads to favorable performance compared to baseline algorithms in terms of either convergence rate or accuracy.

## 2. Background

**Notation.** Matrices are denoted by uppercase letters, such as $A \in \mathbb{C}^{p \times p}$, while lowercase letters, like $b \in \mathbb{C}^p$, represent vectors. Scalars are distinguished based on the context. The Euclidean $\ell_2$-norm is symbolized by $\| \cdot \|_2$. The *qubit* is the basic unit, analogous to the bit in classical computing. A qubit's state is expressed using the Dirac (bra-ket) notation; a single qubit state $|\psi\rangle \in \mathbb{C}^2$ is a linear combination of the basis states $|0\rangle = [1\ 0]^\top$ and $|1\rangle = [0\ 1]^\top$ in $\mathbb{C}^2$. For example, $|\psi\rangle = \alpha|0\rangle + \beta|1\rangle$ where $\alpha, \beta \in \mathbb{C}$ are complex amplitudes. These amplitudes encode the probabilities that the qubit's state collapses to $|0\rangle$ or $|1\rangle$, satisfying the normalization condition $|\alpha|^2 + |\beta|^2 = 1$.

The notation $|\cdot\rangle$ represents a column vector (referred to as a *ket*), and $\langle\cdot|$ denotes its conjugate transpose (*bra*). This is a convenient notation that allows an inner product to be expressed by $\langle x|y\rangle$ and an outer product to be represented by $|x\rangle\langle y|$, with $\langle x|x\rangle = 1$ and $\langle x|y\rangle = 0$, for $x \perp y$. A separable $q$-qubit state, residing in a $q$-qubit Hilbert space, is the Kronecker product of $q$ individual qubit states, represented as $\mathcal{H} = \otimes_{i=1}^q \mathbb{C}^2 \cong \mathbb{C}^{2^q}$. It is expected to equate $2^q = p$. Quantum states are manipulated by quantum gates, which are unitary matrices acting on state vectors. A single-qubit gate $U \in \mathbb{C}^{2 \times 2}$, for example, transforms a state $|\psi\rangle$ into $U|\psi\rangle$, adjusting the state's probability amplitudes according to the operation defined by $U$. An ideal quantum gate $U$ is defined as a unitary matrix, where $U^\dagger U = UU^\dagger = I$ allows rotation effects on $|\psi\rangle$ to preserve the normalization condition of the amplitude.

**Principal component analysis and VQE.** PCA has been studied as a way of finding the representative components from matrix data [36, 37]. This unsupervised learning problem effectively looks for the top (or bottom) principal component of a data matrix calculated via:

$$\max/\min_{v \in \mathbb{R}^p : \|v\|_2 = 1} v^\top M v, \tag{1}$$

where $M = \frac{1}{n} X^\top X \in \mathbb{R}^{p \times p}$ is often the covariance matrix of a dataset $X \in \mathbb{R}^{n \times p}$ with usually centered, normalized rows. For our discussion, assume that $M$ is a real symmetric matrix with eigenvalues and eigenvectors $\{\lambda_i^\star, v_i^\star\}_{i=1}^p$, satisfying $\lambda_1^\star \geq \lambda_2^\star \geq \cdots \geq \lambda_p^\star \geq 0$. Then, $v_1^\star$ corresponds to the vector that maximizes (1) with objective value $v_1^{\star\top} M v_1^\star = \lambda_1^\star$, and $v_p^\star$ is the vector that minimizes (1) with objective value $v_p^{\star\top} M v_p^\star = \lambda_p^\star$.

An interesting aspect of PCA is the connection of (1) with the quadratic form maximization formulations found in the descriptions of quantum problems. In particular, in VQE and given a Hamiltonian $M \in \mathbb{C}^{p \times p}$, the ground state energy, $E_{\min} \in \mathbb{R}$, is always upper-bounded by the expectation of $M$ with respect to a trial wavefunction. That is, $E_{\min} \leq \langle \psi | M | \psi \rangle$ for any $\psi \in \mathbb{C}^p$. Similarly, the maximum energy, $E_{\max} \in \mathbb{R}$, is always lower bounded as in $E_{\max} \geq \langle \psi | M | \psi \rangle$ for any $\psi \in \mathbb{C}^p$.

What is different in VQE is the way we evolve the putative solution: Given an initial point $|\psi_0\rangle$, we start exploring from that point through the dynamics of the variational form $|\psi(\theta)\rangle = \prod_{i=0}^{\ell-1} U_i(\theta_i) |\psi_0\rangle$, where $U_i(\theta_i)$ is a user-defined/designed unitary matrix. In more detail, an ansatz is prepared through a set of parameters $\theta \in \mathbb{R}^m$ to calculate the eigenvector/*eigenstate* corresponding to the largest or least eigenvalue of $M$ via:

$$\max / \min_{\theta \in \mathbb{R}^m} \quad \langle \psi(\theta) | M | \psi(\theta) \rangle \quad \text{s.t.} \quad |\psi(\theta)\rangle = \prod_{i=0}^{\ell-1} U_i(\theta_i) |\psi_0\rangle. \tag{2}$$

Here, the dimension $p = 2^q$ acts on $q$ qubits; $|\psi(\theta)\rangle \in \mathbb{C}^p$ is a quantum state, parameterized by $m$ variables $\theta \in \mathbb{R}^m$; $U_i(\theta_i) \in \mathbb{C}^{p \times p}$ represents a layer of an ansatz, repeated $\times \ell$ times. See Figure 1 for a depiction. Using the bra-ket notation above, the analogs of $i)$ $v \leftrightarrow |\psi(\theta)\rangle$, $ii)$ $x^\top y \leftrightarrow \langle x | y \rangle$, and $iii)$ $\|v\|_2 = 1 \leftrightarrow \||\psi(\theta)\rangle\|_2 = 1$, are determined between PCA and VQE. The key differences are the fact that PCA operates directly on $v$ with no restrictions other than forcing $v$ to be normalized, while, in VQE, we are looking for a parameterized vector $|\psi(\theta)\rangle$ through a specific evolution from an initial state as in $|\psi(\theta)\rangle = \prod_{i=0}^{\ell-1} U_i(\theta_i) |\psi_0\rangle$. Details of the ingredients of (2) are in the text below.

Figure 1: Circuit schematic for VQE where $|\psi(\theta)\rangle$ is prepared as $|\psi(\theta)\rangle = \mathbf{U}(\theta)|\psi_0\rangle$ when $|\psi_0\rangle = |+\rangle$. The initial layer of Hadamard gates may be excluded to have $|\psi_0\rangle = |0\rangle$. The circuit is then measured over an observable $M$ to retrieve $\langle \psi(\theta) | M | \psi(\theta) \rangle$.

*We note that solving the maximization or minimization problem in (1) or (2) is an equivalent problem, since e.g., $\min \langle \psi(\theta) | M | \psi(\theta) \rangle = \max -\langle \psi(\theta) | M | \psi(\theta) \rangle$ by applying a sign operator. For the rest of the text, we will focus on the maximization case.*

---

**Algorithm 1** Ideal VQE with Deflation

---

**Input**: $M \in \mathbb{C}^{p \times p}$, VQE routine for ground state calculation, # of excited states $k$, and iters $t$.

---

$\Psi = \emptyset$
$M_1 \leftarrow M$
**for** $j = 1 : k$ **do**
$\quad |\psi_j(\theta)\rangle \leftarrow \text{VQE}(M_j, t)$ in (7)
$\quad \lambda_j \leftarrow \langle \psi_j(\theta) | M_j | \psi_j(\theta) \rangle$
$\quad M_{j+1} \leftarrow M_j - \lambda_j |\psi_j(\theta)\rangle \langle \psi_j(\theta)|$
$\quad \Psi \leftarrow \Psi \cup \{|\psi_j(\theta)\rangle\}$
**end for**
**Return** Set $\Psi$ of estimated eigenstates.

---

**Beyond "top" eigenstates.** Multicomponent PCA looks for $K$ principal components, which means finding the $K$ most excited states in a VQE setting. For $M \in \mathbb{C}^{p \times p}$ with sorted true principal components $|\psi_1^\star\rangle, |\psi_2^\star\rangle, \ldots, |\psi_p^\star\rangle$, we are interested in finding vectors $|\psi_1\rangle, \ldots, |\psi_k\rangle$, where $k \leq p$, such that the difference between the corresponding true eigenvector $|\psi_i^\star\rangle$ and $|\psi_i\rangle$, for all $i \in [k]$, is bounded as in $\||\psi_i\rangle - |\psi_i^\star\rangle\|_2 \leq \varepsilon$, for some desired bound $\varepsilon$. One way to handle such a case is through deflation [37]: once the largest component $|\psi_1^\star\rangle$ is approximated, $M$ is further processed to "live" on the subspace orthogonal to the subspace spanned by this component. The process then continues by again applying single-component VQE on the deflated $M$, which leads to an approximation of the second component (second excited state) $|\psi_2^\star\rangle$, and so on.

How could quantum deflation look in math? At the $(j+1)$-th iteration of deflation, we estimate the leading eigenvector of:

$$\max_{\theta \in \mathbb{R}^m} \quad \langle \psi(\theta) | M_{j+1} | \psi(\theta) \rangle \quad \text{s.t.} \quad |\psi(\theta)\rangle = \prod_{i=0}^{\ell-1} U_i(\theta_i) |\psi_0\rangle, \tag{3}$$

where $M_{j+1}$ is the deflated Hamiltonian, based on $M_{j+1} = M_j - \lambda_j |\psi_j\rangle\langle\psi_j|$, where $M_j$ is the deflated Hamiltonian in the previous iteration, with $M_1 := M$ and $|\psi_j\rangle$ is the approximate estimate in (7) based on $M_j$ with corresponding energy $\lambda_j := \langle\psi_j|M_j|\psi_j\rangle$. A generic procedure for solving the multicomponent VQE with deflation is detailed in Algorithm 1.

In practice though, successively reconstructing a deflated Hamiltonian represents a computational burden as it would require the construction of the deflated Hamiltonian—that entails a sum of Pauli operators for measuring it. Existing literature relies on implicit deflation algorithms, where one still uses the VQE framework but changes the objective to favor orthogonal solutions by adding a penalty term for vectors that are not orthogonal to the current vector. Such an example is the case of the Variational Quantum Deflation (VQD) algorithm [30]; we describe and compare with VQD in the experimental section.

*In this work, we are exploring a novel penalty term in the objective, inspired by a game-theoretic formulation.*

## 3. An EigenGame for VQE

**What is EigenGame?** A recent study for calculating PCA in a distributed fashion led to the creation of EigenGame [29]. EigenGame is an algorithm posed as a competitive game, where "players" are assigned to estimate eigenvectors, distinct from other eigenvectors of $M$. The way the algorithm operates leads to the Nash equilibrium of an eigenvalue game and does so by a sequential or a parallel version of EigenGame, namely EigenGame and EigenGameR [29], respectively.

Let $v_i \in \mathbb{R}^p$ denote an *approximation* of the $i$-th principal component of $M$. EigenGame defines an alternative *utility objective* to be maximized by the $i$-th "player" in this game, defined as follows:

$$\max_{v_i:\|v_i\|_2=1} \left\{ f(v_i \mid v_{j<i}) := v_i^\top M v_i - \sum_{j<i} \frac{\left(v_i^\top M v_j\right)^2}{v_j^\top M v_j} \right\}. \tag{4}$$

---

**Algorithm 2** EigenGame for the $i$-th player

---

**Input**: $M \in \mathbb{R}^{p \times p}$, initial vectors $v_{i,\text{init}}$, learned parents $v_{j<i}$, step size $\alpha$, and # of iters $t_i$.
$v_{i,1} \leftarrow v_{i,\text{init}}$
**for** $t = 1 : t_i$ **do**

$\quad \nabla_{v_i} f(v_{i,t} \mid v_{j<i}) = 2M\left(v_{i,t} - \sum_{j<i} \frac{v_{i,t}^\top M v_j}{v_j^\top M v_j} v_j\right)$

$\quad \widetilde{v}_{i,t+1} \leftarrow v_{i,t} + \alpha \nabla_{v_i} f(v_{i,t} \mid v_{j<i})$

$\quad v_{i,t+1} \leftarrow \frac{\widetilde{v}_{i,t+1}}{\|\widetilde{v}_{i,t+1}\|_2}$

**end for**
**Return** $v_{i,t_i}$

---

The term in blue maximizes the variance of the projected data along the vector being optimized. This is analogous to the traditional goal of PCA in (1). The term in red penalizes the alignment of the vector with any other vectors that have already been optimized. This term ensures *implicit* orthogonality among the vectors, mimicking the orthogonality constraint in PCA. The intuition behind this redefined objective function is to transform the PCA problem from a passive eigenvalue decomposition into an active, competitive process: each "player"

seeks to maximize its utility in terms of variance captured, considering the presence and position of other vectors. The gradient of the utility function, *assuming access to a first-order oracle*, can be calculated as $\nabla_{v_i} f(v_i \mid v_{j<i}) = 2M\left(v_i - \sum_{j<i} \frac{v_i^\top M v_j}{v_j^\top M v_j} v_j\right)$. We describe EigenGame in Algorithm 2.

**Preparing the quantum EigenGame.** Our interest steers towards obtaining a utility gradient that uses quantum oracle calls to calculate eigenvalue/eigenvector pairs. We first convert the classical EigenGame objective formulation into a parameterized one, using quantum computing formulations. Denoting the $r$-th player's parameters as $\theta^{(r)} \in \mathbb{R}^m$, its corresponding objective becomes:

$$\max_{\theta^{(r)} \in \mathbb{R}^m} \left\{ \langle\psi(\theta^{(r)})|M|\psi(\theta^{(r)})\rangle - \sum_{j<r} \frac{\langle\psi(\theta^{(r)})|M|\psi(\theta^{(j)})\rangle^2}{\langle\psi(\theta^{(j)})|M|\psi(\theta^{(j)})\rangle} \right\} \quad \text{s.t.} \quad |\psi(\theta^{(r)})\rangle = \prod_{i=0}^{\ell-1} U_i(\theta_i^{(r)})|\psi_0\rangle. \tag{5}$$

What differentiates (5) from the classical VQE formulation in (2) is the inclusion of the regularization term in red. That is, (5) defines different objectives per "player". Note that the term in blue is

the measurement of the observable Hamiltonian $M$ for the $r$-th player and is identical to that of the VQE objective in (2).

Focusing on the red part in (5), to incorporate information from the previous eigenvalue/eigenvector pairs, we compute the term $\langle\psi(\theta^{(r)})|M|\psi(\theta^{(j)})\rangle^2$ using a procedure that calculates inner products on quantum states. We do so through an approach similar to the `Hadamard Test` [38, 39], where we first create an equal superposition of $|\psi(\theta^{(j)})\rangle$ and $|\psi(\theta^{(r)})\rangle$ with one additional ancilla qubit, followed by an $H$ gate on said ancilla to yield $\mathrm{Re}(\langle\psi(\theta^{(r)})|M|\psi(\theta^{(j)})\rangle)$ via Hamiltonian expectation with $M = R(\omega)^\dagger Z R(\omega)$. By adding an $S$ gate, we can compute $\mathrm{Im}(\langle\psi(\theta^{(r)})|M|\psi(\theta^{(j)})\rangle)$. Here, $H = \frac{1}{\sqrt{2}}\begin{bmatrix} 1 & 1 \\ 1 & -1 \end{bmatrix}$ and $S = \begin{bmatrix} 1 & 0 \\ 0 & i \end{bmatrix}$ are single-qubit uni-

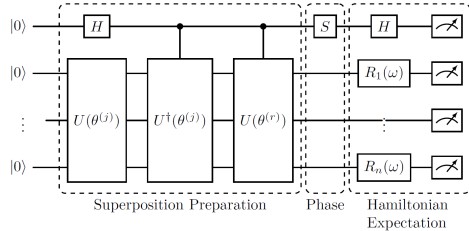

Figure 2: To compute terms of the form $\langle\psi(\theta^{(r)})|M|\psi(\theta^{(j)})\rangle$, we utilize interference between $|\psi(\theta^{(r)})\rangle$ and $|\psi(\theta^{(j)})\rangle$.

tary gates, $Z = \begin{bmatrix} 1 & 0 \\ 0 & -1 \end{bmatrix}$ is the computational basis Hamiltonian, and $R(\omega)$ is an arbitrary rotation gate. The ancilla qubit is an auxiliary qubit useful for intermediate computations. For an ancilla qubit with state $|a\rangle$, the state of an n-qubit system that contains an ancilla can be defined as $|\Psi\rangle = \sum_{i=0}^{2^n-1} \alpha_i|i\rangle \otimes |a\rangle$. This implementation shown in Figure 2 computes $\langle\psi(\theta^{(r)})|M|\psi(\theta^{(j)})\rangle$ with only one additional qubit and two quantum oracle calls, providing an effective solution. See Appendix B for a detailed discussion and a derivation for our implementation. From here, we will also refer to quantum EigenGame as QuantumGame.

---

**Algorithm 3** `QuantumGame` of $r$-th player

---

**Input**: matrix $M \in \mathbb{C}^{p\times p}$, initial values $\theta_{\mathrm{init}}^{(r)} \in \mathbb{R}^m$, # of iters/ $T$, # of shots $S$, step size $\eta > 0$.

$\theta_{1,:}^{(r)} := \theta_{\mathrm{init}}^{(r)}$.
**for** $t = 1 : T$ **do**
  Prepare state $|\psi(\theta_{t,:}^{(r)})\rangle = \prod_{i=0}^{\ell-1} U_i(\theta_{t,i}^{(r)})|+\rangle$.
  Use $|\psi(\theta_{t,:}^{(r)})\rangle$ to approximate the gradient of objective in (5) w.r.t. $\theta^{(r)}$; denote as $\widetilde{\nabla}_{t,:}^{(r)} \in \mathbb{R}^m$.
  Update $\theta_{t+1,:}^{(r)} = \theta_{t,:}^{(r)} + \eta\widetilde{\nabla}_{t,:}^{(r)}$.
**end for**
**Return** $|\psi(\theta^{(r)})\rangle := |\psi(\theta_{T+1,:}^{(r)})\rangle$.

---

Based on the above, Algorithm 3 provides our algorithm for the $r$-th player. Note the abuse of notation where $\theta_{t,i}^{(r)}$ indicates the variational parameters of the $r$-th player, associated with the $i$-th layer of the VQE that is $\subset \theta_{t,:}^{(r)}$ at the $t$-th iteration. Then, given the Hamiltonian $M$, and an initialization $\theta_{\mathrm{init}}^{(r)} \in \mathbb{R}^m$ for the variational parameters, Algorithm 3 repeats over $T$ iterations the steps of: $i)$ preparing the quantum state $|\psi(\theta_{t,:}^{(r)})\rangle = \prod_{i=0}^{\ell-1} U_i(\theta_{t,i}^{(r)})|+\rangle$; $ii)$ calculating gradient approximation of the objective in (5) w.r.t. $\theta_{t,:}^{(r)}$, and $iii)$ performing gradient ascent on the classical side (for the case of maximization) with step size $\eta$ to update $\theta^{(r)}$.

At the end of the execution, the eigenvalue $\langle\psi(\theta^{(r)})|M|\psi(\theta^{(r)})\rangle$ is broadcast to all "children" agents $r' > r$, alongside with the corresponding eigenvector $|\psi(\theta^{(r)})\rangle$ to form (5) for the next "player". This process has a distributed flavor where each eigenvalue/eigenvector pair a parent agent calculates is broadcast to all its children agents, creating a directed graph acyclic hierarchy; see also Figure 3.

**Challenges for the quantum EigenGame.** Quantum computers typically interact with $0^{\mathrm{th}}$-order oracles, meaning they can only access function values without gradient information. This limitation impacts the efficiency and precision of algorithms, like EigenGame, which rely on gradients [29].

Any oracle call on a quantum computer is inherently noisy. The noise arises from multiple sources, including the underlying Hamiltonian, the calibration of quantum gates, cross-talk, the accuracy of the measurement mechanisms, and stochastic errors associated with measurements [4–6].

---

[2]Note that the term $\langle\psi(\theta^{(j)})|M|\psi(\theta^{(j)})\rangle$ in the denominator of the red term is a classical observable operation, given the prepared state $|\psi(\theta^{(j)})\rangle$.

Finally, to our knowledge, the convergence analysis of the EigenGame under the effect of errors is not well understood. Traditional convergence proofs assume ideal conditions with precise computations, which do not hold in practical quantum scenarios. Recent advances in quantum algorithm analysis provide a foundation, but specific adaptations for the EigenGame are necessary [19].

Regarding gradient approximation, recent works leverage *parameter-shift* rules [38, 40–43] to obtain exact gradients using the same number of oracle evaluations as finite differences, given prior knowledge on the eigenvalues of the applied ansatz. In particular, for a single-qubit unitary with two distinct eigenvalues $\pm\lambda$, the exact derivative is: $\frac{\partial f(\theta)}{\partial_m} = \lambda \left[ f\left(\theta + \frac{\pi}{4\lambda}\hat{e}_m\right) - f\left(\theta - \frac{\pi}{4\lambda}\hat{e}_m\right) \right]$ [38].

**Classical $0^{\text{th}}$-order EigenGame.** To better understand the nature of having $0^{\text{th}}$-order oracles in QuantumGame, we analyze the classical Eigengame with that same constraint. In principle, the $0^{\text{th}}$-order version of EigenGame should not require gradient information, relying instead on function evaluations to guide the search for optimal solutions. These techniques, such as forward finite differences [44] and simultaneous perturbation stochastic approximation (SPSA) [45], provide approximate gradient information through multiple function evaluations [18].

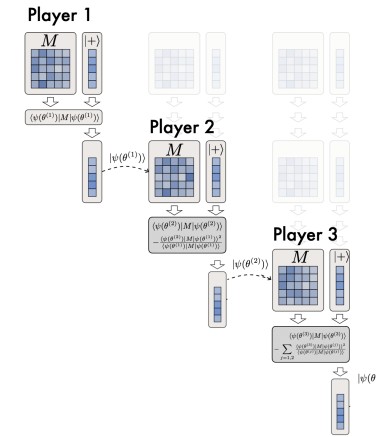

Figure 3: QuantumGame with three players.

For simplicity, we will focus on the *forward finite differences* technique. Based on the objective in (4), we study the case when an error $\sigma \in \mathbb{R}$ appears in each partial derivative. The error $\sigma$ can be purely stochastic or predefined, and the treatment we give to it will differ based on its nature. Then, the $0^{\text{th}}$-order utility partial derivative with respect to the $m$-th component of $v_i \in \mathbb{R}^p$ can be approximated as:

$$\frac{\partial f(v_i \mid v_{j<i})}{\partial v_{i,m}} \approx \frac{f(v_{i,1}, \ldots, v_{i,m} + \sigma, \ldots, v_{i,p} \mid v_{j<i})) - f(v_i \mid v_{j<i}))}{\sigma}$$

The above lead to the following observation, that will be helpful in the theory we will present:

**Observation 1** *We perform forward finite differences in order to obtain an analytical expression for a $\sigma$-approximate partial derivative of the $m$-th component of $v_i$, which yields (see Appendix A):*

$$\widetilde{\nabla}_{v_i} f(v_i \mid v_{j<i}) = 2M\left(v_i - \sum_{j<i} \frac{v_i^\top M v_j}{v_j^\top M v_j} v_j\right) + \sigma\left(diag(M) - \sum_{j<i} \frac{(Mv_j)^{\circ 2}}{v_j^\top M v_j}\right). \qquad (6)$$

---

**Algorithm 4** $0^{\text{th}}$-order EigenGame for $i$-th player

---

**Input**: $M \in \mathbb{R}^{p\times p}$, initial vectors $v_{i,\text{init}}$, learned approximate parents $v_{j<i}$, perturbation $\sigma \in \mathbb{R}$ and step size $\alpha$, # of iters. $t_i$

$v_{i,1} \leftarrow v_{i,\text{init}}$
**for** $t = 1 : t_i$ **do**

$\quad \texttt{utility} := 2M\left(v_{i,t} - \sum_{j<i} \frac{v_{i,t}^\top M v_j}{v_j^\top M v_j} v_j\right)$

$\quad \texttt{error} := \sigma\left(\texttt{diag}(M) - \sum_{j<i} \frac{(Mv_j)^{\circ 2}}{v_j^\top M v_j}\right)$

$\quad \widetilde{\nabla}_{v_i} f(v_{i,t} \mid v_{j<i}) = \texttt{utility} + \texttt{error}$

$\quad \widetilde{v}_{i,t+1} \leftarrow v_{i,t} + \alpha \widetilde{\nabla}_{v_i} f(v_{i,t} \mid v_{j<i})$

$\quad v_{i,t+1} \leftarrow \frac{\widetilde{v}_{i,t+1}}{\|\widetilde{v}_{i,t+1}\|_2}$

**end for**
**Return** $v_{i,t_i}$

---

Observe a linear dependence on $\sigma$ on the second term, while the first term is identical to the analytical utility gradient. Based on (6), Algorithm 4 describes the $0^{\text{th}}$-order EigenGame procedure including both the analytical and the error terms of the *finite differences* approximated utility gradient.

**Theoretical guarantees.** We provide convergence proofs for the $0^{\text{th}}$-order EigenGame and for the parameterized EigenGame in Appendices C and E, respectively, and error accumulation theory for both in Appendix G. We first summarize the global convergence rate for both algorithms in the following.

**Theorem 1** (*Convergence of $0^{\text{th}}$-order EigenGame for all players*). *Consider the Algorithm 4 with input matrix $M \in \mathbb{R}^{p\times p}$ and learned*

*"parent" eigenvectors $v_{j<i} \in \mathbb{R}^p$ that are accurate enough, i.e., that $|\phi_{j<i}| \leq \frac{c_i g_i}{(i-1)\Lambda_{11}} \leq \sqrt{\frac{1}{2}}$ with $0 \leq c_i \leq \frac{1}{16}$. Let the initialization vector $v_{i,init}$ be within perturbation $\frac{\pi}{4}$ from $v_i^\star$, i.e., $\angle(v_{i,init}, v_i^\star) \leq \frac{\pi}{4}$, for all $i$. Consider perturbation $\sigma \in \mathbb{R}$ for the finite difference approximation, and step size $\alpha$ for the gradient ascent. Then, Algorithm 4 returns an approximate eigenvector $v_i$ with angular error less than $\phi_{tol} > 0$ in*

$$T = \left\lceil \mathcal{O}\left( \sum_{i=1}^{k} \frac{L_i(0)}{L_i(\sigma)} \left[ \frac{(k-1)!}{\phi_{tol}} \prod_{j=1}^{k} \left( \frac{16\Lambda_{11}}{g_j} \right) \right]^2 \right) \right\rceil \quad \text{iterations,}$$

*where $L_i(\sigma)$ is the Lipschitz continuity assumption of the $0^{th}$-order EigenGame based on a finite difference step size $\sigma$ as in $\|\widetilde{\nabla}_{v_i} f(v_i \mid v_{j<i})\|_2 \leq L_i(\sigma)$, $\Lambda$ is the diagonal eigenvalue matrix of $M$ containing eigenvalues $\Lambda_{11} > \Lambda_{22} > \ldots > \Lambda_{kk}$ with $\Lambda_{11}$ being the top eigenvalue, and $g_i = \Lambda_{ii} - \Lambda_{i+1,i+1}$ is the eigengap between the two consecutive eigenvalues of players $i$ and $i + 1$. The proof of this theorem is developed in Appendix D.*

**Theorem 2** (*Convergence of QuantumGame for all players*). *Under sufficient decrease assumptions, Algorithm 3 achieves convergence to within $\phi_{tol}$ angular error of the top-k principal components independent of initialization. Let $|\phi_{j<i}| \leq \frac{c_i g_i}{(i-1)\Lambda_{11}} \leq \sqrt{\frac{1}{2}}$ with $0 \leq c_i \leq \frac{1}{16}$. Let each $v(\hat{\theta}_i) = U(\hat{\theta}_i)|s\rangle$, with a sufficiently expressive ansatz $U(\hat{\theta}_i)$ [46, 47] and an initial state $|s\rangle$ such that $\angle(v(\hat{\theta}_i), v_i^\star) \leq \frac{\pi}{4}$. Algorithm 3 returns the eigenvectors with angular error less than $\phi_{tol}$ in*

$$T = \left\lceil \mathcal{O}\left( \sum_{i=1}^{k} \frac{L_{\theta_i}{}^2}{\sqrt{\ell q}} \left[ \frac{(k-1)!}{\phi_{tol}} \prod_{j=1}^{k} \left( \frac{16\Lambda_{11}}{g_j} \right) \right]^2 \right) \right\rceil \quad \text{iterations,}$$

*where $L_{\theta_i}$ is the Lipschitz continuity constant of QuantumGame, $\ell$ is the number of layers of the ansatz and $q$ is the number of qubits. The proof is shown in Appendix F.*

We are also interested in establishing the error accumulation rate in both algorithms, given inaccurate parents. To do so, we bound the gradient difference between exact and inexact parent eigenvectors. Our error accumulation theorem is stated as follows.

**Theorem 3** ($0^{th}$-*order and parameterized error accumulation*). *Assume the angular error $\phi_j \leq \epsilon$ between the true eigenvector $v_j$, $j < i$ of a parent and its estimate $\hat{v}_j$ to satisfy $\epsilon \ll 1$. The Euclidean error of the parent is $\mathcal{O}(\epsilon)$ and the Euclidean error of the child's $0^{th}$-**order** gradient is $\mathcal{O}(\epsilon)$. Similarly, the Euclidean error of the child's **parameterized** gradient is $\mathcal{O}(\epsilon\sqrt{\ell q})$, with $\ell$ being the number of layers and $q$ the number of qubits of the parameter space. An illustration for the $0^{th}$-order case is shown in Figure 4, and proofs are given in Appendix G.*

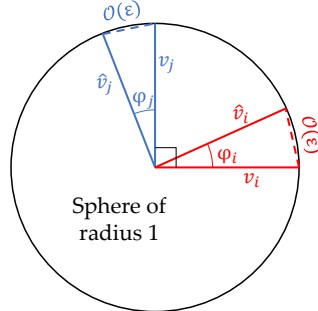

Figure 4: Error accumulation of a child eigenvector $\hat{v}_i$ from its parent eigenvector $\hat{v}_j$, given $\phi_j \leq \epsilon \ll 1$.

*Proof Sketch of Theorems 1 and 2.* The $0^{th}$-order theory begins by specifying the assumptions of *utility lower bound* (Assumption 1) and *sufficient decrease* (Assumption 2) necessary to identify the finite sample convergence rate of the $i$-th player of Algorithm 4. Theorem 5 specifies the generic Riemannian descent convergence rate under both assumptions, which we use for Riemannian ascent instead. The theory follows by defining the conditions under which the assumptions are satisfied, and used in Theorem 5.

Lemmas 9 and 10 provide a Lipschitz continuity coefficient $L_i(\sigma)$ that admits a finite-differences approximation error $\sigma$ for the $i$-th player, where $L_i(0)$ becomes the Lipschitz coefficient of the exact EigenGame gradient. Corollary 11 approximately bounds the utility using $L_i(\sigma)$, approximately satisfying Assumption 1. Lemma 12 then incorporates Corollary 11 to approximately bound the difference in the utility of consecutive steps, approximately satisfying Assumption 2. We then input these values into Theorem 5, and thus determine the number of iterations required for the $0^{th}$-order EigenGame to converge in Lemma 15.

We employ a similar strategy for the parameterized scenario that we implement on quantum devices: we aim to satisfy Assumptions 1 and 2, and find the convergence rate for each player. To that effect, we assume the use of a highly expressive ansatz that can reach the exact eigenvalue of interest, and include information on the number of parameters of the ansatz in our analysis.

Lemmas 17 and 18 determine a Lipschitz continuity constant $L_{\hat{\phi}_i}$ for the $i$-th player, introducing the number of layers $\ell$ and the number of qubits $q$ from the ansatz. Corollary 19 states that the utility of the parameterized EigenGame is bounded by the same $L_i$ constant of the original EigenGame, satisying Assumption 1. We use a first-order Taylor expansion of the utility to approximately satisfy Assumption 2 in Lemma 20. The identified values that satisfy Assumption 1 and approximately satisfy Assumption 2 are used to provide the number of iterations necessary to reach finite sample convergence for QuantumGame in Lemma 22.

# 4. Experiments and discussion

We are interested in understanding how well our versions of EigenGame perform compared to the state of the art. With the application being the retrieval of energy levels for specific molecules, our experiments focus on characterizing the energy levels of the $H_2$ molecule, as well as understanding the impact of noise on QuantumGame compared to the Variational Quantum Deflation (VQD) algorithm.

**Experimental setup.** We use Pennylane's `RandomLayers` ansatz [48] with a number of layers $\ell$ proportional to the problem size of $q$ qubits, following ansatz design suggestions from [49] where a direct relation between the degree of entanglement and the number of layers of an ansatz is made in the low-qubit regime, showing a linear scaling for layers that we adopt here. We study a 2-qubit version of the $H_2$ molecule by applying a `ParityMapper` to the second quantization Hamiltonian of $H_2$. We select a low number of 3 parameters per layer with 3 layers, sufficient to obtain the correct eigenvalues in a noiseless scenario.

**Baselines.** We compare results against the Variational Quantum Deflation (VQD) algorithm [30], which aims to optimize:

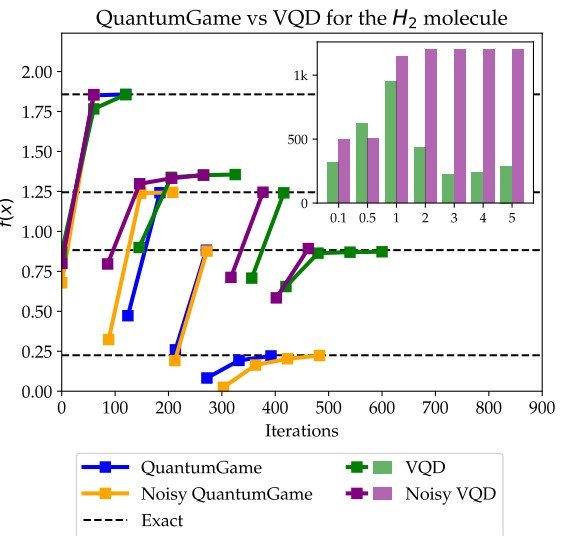

Figure 5: Excited state energy levels of the two-qubit $H_2$ molecule mapping calculated with noiseless QuantumGame (blue line), noisy QuantumGame (orange line), noiseless VQD (green line) and noisy VQD (purple line) over accumulative iterations. The noisy setting only uses statistical shot noise with 10k shots. The exact energy level (dashed line) is extracted using Numpy. The inset plot shows the total number of iterations for noiseless VQD and noisy VQD with values of $\beta \in [0.1, 0.5, 1, 2, 3, 4, 5]$.

$$\min_{\theta^{(r)} \in \mathbb{R}^m} \quad \langle \psi(\theta^{(r)})|M|\psi(\theta^{(r)})\rangle + \sum_{j<r} \beta |\langle \psi(\theta^{(r)})|\psi(\theta^{(j)})\rangle|^2 \quad \text{s.t.} \quad |\psi(\theta^{(r)})\rangle = \prod_{i=0}^{\ell-1} U_i(\theta_i^{(r)})|\psi_0\rangle, \quad (7)$$

a similar approach to our algorithm but instead including a regularization parameter $\beta$ which requires adequate tuning to reach comparable accuracy, adding a small overhead. We avoid this overhead by using an energy-aware adaptive regularization term, scaled with previously calculated eigenvalues. We test VQD over $\beta \in [0.1, 0.5, 1, 2, 3, 4, 5]$ and set it to $0.1$ in the main plot of Figure 5, with the least total number of iterations among the options tested. The total number of iterations for all $\beta$ values are shown in the inset plot of Figure 5. Our settings for QuantumGame and VQD experiments are comprised of exact statevector simulation and shot-based simulation, where results are estimated based on averaging values over $10k$ shots or samples affected by statistical shot noise through the expression $\langle M \rangle_{\text{est}} \approx \langle M \rangle \pm \sqrt{\frac{\text{Var}(M)}{N}}$, where $N$ is the number of shots, $\langle M \rangle_{\text{est}}$ is the estimate of the expectation value of $M$ and $\text{Var}(M) = \langle M^2 \rangle - \langle M \rangle^2$ is its variance [48].

**Results.** Figure 5 illustrates the convergence of the QuantumGame and VQD to the energy levels of the $H_2$ molecule when no noise effects are present and when under the effect of statistical shot noise –our noisy scenario– for 10k shots. We use the `Gradient Descent` optimizer with $\eta = 1/2L$, $L = \|M\|$, and a tolerance $\|\nabla_{\hat{\theta}_i} f(v(\hat{\theta}_i), v(\hat{\theta}_{j<i}))\|_2 \leq 10^{-2}$. A similar convergence rate can be observed for the eigenvalue estimates of VQD when noise is present against their noiseless counterparts for $\beta \leq 1$, with shot noise being a larger source of errors among all values of $\beta$ tested. This could be caused by our choices of the `Gradient Descent` optimizer, step size $\eta$, or the number of shots. For any of the scenarios, either more parameters or more shots may aid in obtaining higher accuracy. QuantumGame thus shows more accurate results than the baseline without hyperparameter tuning.

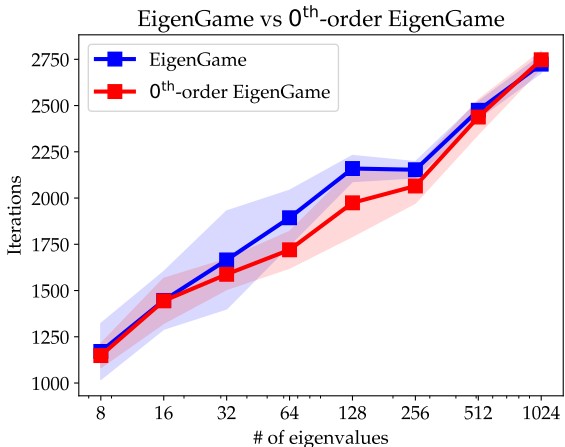

Figure 6: Total number of iterations required for eigenvalue calculation to within a tolerance of $\rho_i \leq 10^{-3}$, $i < k$ using the exact and the $0^{\text{th}}$-order EigenGame over $k = \{2^i \mid i \in [3, 10]\}$ non-degenerate eigenvalues sampled from a power-law distribution over 5 runs. The blue and red line correspond to the exact EigenGame and the $0^{\text{th}}$-order EigenGame, respectively.

**Ablation studies.** We also inspect the behavior of our $0^{\text{th}}$-order EigenGame compared to the exact EigenGame to validate the accuracy and convergence rate of the algorithms. These experiments are performed in a classical noiseless setting, as mapping exact statevectors $v_i$ onto a quantum computer is impractical given the exponential resources required. We first prepare the $M$ Hamiltonian matrix from which we calculate the 8 leading eigenvalue/eigenvector pairs. Our focus is on finding the number of iterations required to reach $\|\nabla_{v_i} f(v_i \mid v_{j<i})\|_2 \leq 10^{-3}$ on all $i < k$ eigenvalues, for $k = \{2^i \mid i \in [3, 10]\}$. In order to have adequate control over the optimization process and to ensure results that don't end up as corner cases, we construct $M$ using a set of eigenvalues sampled from a power-law distribution in the range $(0, 1)$ and employing a similarity transformation through $M = P^{-1}DP$ with $D = \lambda I$ a diagonal matrix with eigenvalues on the diagonal, and $\lambda$ being a set of non-negative and non-degenerate eigenvalues. Given no prior knowledge on the eigenvectors to calculate, we generate a random invertible matrix $P$ as a random orthonormal matrix through a $QR$ decomposition where $Q = P$, $P^\dagger = P^{-1}$, $P \in \mathbb{C}^{n \times n}$. Figure 6 describes our results and shows a similar scaling trend on both algorithms when using the `Gradient Descent` optimizer with $\eta = 1/2L$, $L = \|M\|$.

# 5. Conclusions

We propose a variational algorithm for the problem of finding the top $k$ eigenvalues of a Hamiltonian based on EigenGame [50]. We also propose a $0^{\text{th}}$-order EigenGame using forward finite differences based on a $0^{\text{th}}$-order oracle restriction motivated by the oracle class available to quantum computers. We perform a conventional classical convergence analysis for the $0^{\text{th}}$-order EigenGame and state a global convergence rate, along with an error accumulation rate that accounts for imprecise parents. A similar analysis is performed for QuantumGame, where we include the number of layers and qubits of an ansatz.

We conduct an ablation study on the $0^{\text{th}}$-order EigenGame by comparing the number of steps required to accurately estimate the top $k$ eigenvalues of a Hamiltonian $M$ against the exact EigenGame for small, medium and large dimensionality. We observe a similar scaling trend, implying that despite numerical errors being introduced, correct eigenvalues can still be found. We compared our QuantumGame formulation against the VQD algorithm and found an advantage of our formulation over VQD in experimental convergence rate and accuracy.

**Future work.** A parallel implementation of QuantumGame where eigenvalues are calculated on different quantum devices simultaneously. It is our aim to remove any unnecessary assumptions in the analysis of our algorithms to strengthen our theory.

## Acknowledgements

This research was funded in part by: The Robert A. Welch Foundation (grant No. C-2118 A.K.); Rice University (Faculty Initiative award); NSF FET:Small (award no. 1907936); NSF CMMI (award no. 2037545); NSF CAREER (award no. 2145629); a Rice InterDisciplinary Excellence Award (IDEA); an Amazon Research Award; a Microsoft Research Award. AK would also like to thank the Ken Kennedy Institute for its support through the Research Cluster "QuanTAS". The content is solely the responsibility of the authors and does not necessarily represent the official views of the Funders. We thank professors Cesar Uribe and Tirthak Patel for insightful discussions on the project.

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

# A. Calculating the expression of the finite differences EigenGame gradient

To map the finite differences formulation on the EigenGame objective, we delve into the specifics of the utility function, by unrolling the matrix/vector inner products entrywise:

$$f(v_i \mid v_{j<i}) = v_i^\top M v_i - \sum_{j<i} \frac{(v_i^\top M v_j)^2}{v_j^\top M v_j}$$

$$= \sum_l \sum_k v_{i,l} v_{i,k} M_{l,k} - \sum_{j<i} \frac{(\sum_l \sum_k v_{j,l} v_{i,k} M_{k,l})^2}{\sum_l \sum_k v_{j,l} v_{j,k} M_{l,k}}$$

$$= v_{i,m} \cdot \left( \sum_{k\neq m} v_{i,k} M_{m,k} \right) + \left( \sum_{k\neq m} v_{i,k} M_{k,m} \right) \cdot v_{i,m} + v_{i,m}^2 M_{m,m} + \sum_{l\neq m} \sum_{k\neq m} v_{i,l} v_{i,k} M_{l,k}$$

$$- \sum_{j<i} \frac{\left( v_{i,m} \cdot (\sum_l M_{m,l} v_{j,l}) + \sum_l \sum_{k\neq m} v_{j,l} v_{i,k} M_{k,l} \right)^2}{v_j^\top M v_j}$$

where blue terms correspond to the components from the core objective term $v_i^\top M v_i$, while red terms correspond to the regularized term of the EigenGame formulation, $\sum_{j<i} \frac{(v_i^\top M v_j)^2}{v_j^\top M v_j}$. Adding the error term $\sigma$ to the $m$-entry of $v_i$, we obtain:

$$f(v_{i,1}, \ldots, v_{i,m} + \sigma, \ldots, v_{i,p} \mid v_{j<i}) = (v_{i,m} + \sigma) \cdot \left( \sum_{k\neq m} v_{i,k} M_{m,k} \right) + \left( \sum_{k\neq m} v_{i,k} M_{k,m} \right)$$

$$\cdot (v_{i,m} + \sigma) + (v_{i,m} + \sigma)^2 M_{m,m} + \sum_{l\neq m} \sum_{k\neq m} v_{i,l} v_{i,k} M_{l,k}$$

$$- \sum_{j<i} \frac{\left( (v_{i,m} + \sigma) \cdot (\sum_l M_{m,l} v_{j,l}) + \sum_l \sum_{k\neq m} v_{j,l} v_{i,k} M_{k,l} \right)^2}{v_j^\top M v_j}$$

Next, we perform *forward finite differences* in order to obtain an expression for an $\sigma$-approximate partial derivative of the $m$-th component of $v_i$:

$$\frac{f(v_{i,1}, \ldots, v_{i,m} + \sigma, \ldots, v_{i,p} \mid v_{j<i}) - f(v_i \mid v_{j<i})}{\sigma}$$

$$= \frac{\sigma}{\sigma} \cdot \left( \sum_{k\neq m} v_{i,k} M_{m,k} + \sum_{k\neq m} v_{i,k} M_{k,m} + (2v_{i,m} + \sigma) M_{m,m} \right.$$

$$\left. - \sum_{j<i} \frac{(2v_{i,m} + \sigma)(\sum_l M_{m,l} v_{j,l})^2}{v_j^\top M v_j} - \sum_{j<i} \frac{2(\sum_l M_{m,l} v_{j,l})(\sum_l \sum_{k\neq m} v_{j,l} v_{i,k} M_{k,l})}{v_j^\top M v_j} \right)$$

$$= 2 M_{m,:} v_i + \sigma M_{m,m} - \sum_{j<i} \left( \frac{2 M_{m,:} \cdot v_j \cdot (v_i^\top M v_j) + \sigma \cdot (M_{m,:} v_j) \cdot (M_{m,:} v_j)}{v_j^\top M v_j} \right)$$

$$= 2 M_{m,:} \left( v_i - \sum_{j<i} \frac{v_i^\top M v_j}{v_j^\top M v_j} v_j \right) + \sigma \left( M_{m,m} - \sum_{j<i} \frac{(M_{m,:} v_j)^2}{v_j^\top M v_j} \right)$$

Finally, the previous expression, in a vectorized gradient form, is equivalent to:

$$\widetilde{\nabla}_{v_i} f(v_i \mid v_{j<i}) = 2M \left( v_i - \sum_{j<i} \frac{v_i^\top M v_j}{v_j^\top M v_j} v_j \right) + \sigma \left( \mathtt{diag}(M) - \sum_{j<i} \frac{(Mv_j)^{\circ 2}}{v_j^\top M v_j} \right). \tag{8}$$

## B. Swap test and Hadamard test derivations

The `SwapTest` [51, 52] calculates the inner product between two quantum states. It requires $2q + 1$ qubits for its operation, including $q$ qubits for representing each of both quantum states, and 1 ancilla qubit that will store the inner product. In particular, for two states $|\phi_1\rangle, |\phi_2\rangle \in \mathbb{C}^p$, the `SwapTest` circuit outputs a probability value:

$$P(|0\ldots\rangle) = \tfrac{1}{2} - \tfrac{1}{2}|\langle\phi_1|\phi_2\rangle|^2 \in [1/2, 1], \tag{9}$$

that represents the probability of the first qubit in a quantum state being in the $|0\rangle$ state. E.g., when $|\phi_1\rangle, |\phi_2\rangle$ are coaligned, then $|\langle\phi_1|\phi_2\rangle| = 1$, and thus $P(|0\ldots\rangle) = 1$, while when $|\phi_1\rangle, |\phi_2\rangle$ are orthogonal, then $|\langle\phi_1|\phi_2\rangle| = 0$ and thus $P(|0\ldots\rangle) = \tfrac{1}{2}$ (i.e., observing 0 is equivalent to a toss of a perfect coin). This allows one to, for example, estimate the squared inner product between the two states, $|\langle\phi_1|\phi_2\rangle|^2$, to $\varepsilon$ additive error by taking the average over $O(\tfrac{1}{\varepsilon^2})$ runs of the `SwapTest`.

Figure 7 shows our implementation of the procedure, where we prepare:

$$|\phi_1\rangle = |\psi(\theta^{(i)})\rangle, \quad |\phi_2\rangle = M|\psi(\theta^{(j)})\rangle,$$

for each index in the sum of eq. 5, obtaining:

$$|\langle\psi(\theta^{(i)})|M|\psi(\theta^{(j)})\rangle|^2 = 1 - 2P(|0\ldots\rangle). \tag{10}$$

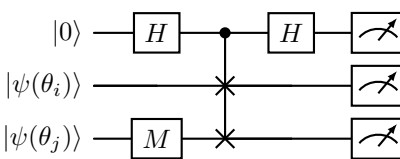

Figure 7: Circuit schematic for SwapTest. Here, $M$ is the problem Hamiltonian, and $H$ is the Hadamard gate.

In order to correctly implement the SwapTest, the matrix $M$ must satisfy one of the following requirements:

- $M$ is a unitary matrix.
- $M$ can be decomposed into a linear combination of Pauli operators $\{P_i\}$ as $M = \sum_i a_i P_i$ such that each evaluation $\langle\hat{v}_i(\theta_i)|P_i|\hat{v}_j(\theta_i)\rangle^2 \geq 0$.
- $M$ can be decomposed into a linear combination of unitary matrices $\{U_i\}$ as $M = \sum_i a_i U_i$ such that each evaluation $\langle\hat{v}_i(\theta_i)|U_i|\hat{v}_j(\theta_i)\rangle^2 \geq 0$.

These limitations are due to the `SwapTest` being only able to calculate the overlap of quantum states when $M$ is a quantum gate applied to one of the two states. If $M$ is already a unitary matrix, only one evaluation is necessary. On the other hand, if $M$ is not a unitary matrix, it should be decomposed into a linear combination of unitary matrices such as Pauli operators. After a successful decomposition of $M$ as $M = \sum_i a_i U_i$, the following quantities may be calculated:

$$|\langle\psi(\theta_i)|U_i|\psi(\theta_j)\rangle| = \sqrt{1 - 2P_i(|0\ldots\rangle)}. \tag{11}$$

which can be combined and contrasted with the expected calculation via

$$\begin{aligned} |\langle\psi(\theta_i)|M|\psi(\theta_j)\rangle| &= \left| \sum_i a_i \langle\psi(\theta_i)|U_i|\psi(\theta_j)\rangle \right| \\ &\geq \sum_i a_i |\langle\psi(\theta_i)|U_i|\psi(\theta_j)\rangle|. \end{aligned} \tag{12}$$

In order to reach an equivalence, either $M = U_0$ or:

$$\langle\psi(\theta_i)|U_i|\psi(\theta_j)\rangle \geq 0, a_i > 0 \tag{13}$$

must be satisfied.

In order to overcome the limitation of (13), we propose a novel circuit for computing this value, illustrated in Figure 2. We will describe how this implementation computes $\langle\psi(\theta^{(r)})|M|\psi(\theta^{(j)})\rangle$ via the following Lemma:

**Lemma 4** *The circuit depicted in Figure 2 is able to compute the value $\langle\psi(\theta^{(r)})|M|\psi(\theta^{(j)})\rangle$ using $q + 1$ qubits and two expectations of $M$.*

*Proof.* Without loss of generality, assume we are computing $\text{Re}(\langle\psi(\theta^{(r)})|M|\psi(\theta^{(j)})\rangle)$ (as computing $\text{Im}(\langle\psi(\theta^{(r)})|M|\psi(\theta^{(j)})\rangle)$ is an analogous operation). We will show that we can compute $\text{Re}(\langle\psi(\theta^{(r)})|M|\psi(\theta^{(j)})\rangle)$ with $q+1$ qubits and one expectation of $M$.

Let $|\psi(\theta^{(j)})\rangle = U(\theta^{(j)})|0\rangle$ and $|\psi(\theta^{(r)})\rangle = U(\theta^{(r)})|0\rangle$, after preparing the superposition in Figure 2, if we let $\boldsymbol{\Psi}$ represent the state of our quantum system, then $\boldsymbol{\Psi}$ can be written as:

$$\boldsymbol{\Psi} = \frac{1}{\sqrt{2}}(|\psi(\theta^{(j)})\rangle|0\rangle + |\psi(\theta^{(r)})\rangle|1\rangle) \tag{14}$$

We then apply the phase gate $S$ to $\boldsymbol{\Psi}$ depending on whether we want to measure the imaginary component of $\langle\psi(\theta^{(r)})|\hat{H}|\psi(\theta^{(j)})\rangle$. We will proceed our analysis without applying the $S$ gate to compute $\text{Re}(\langle\psi(\theta^{(r)})|\hat{H}|\psi(\theta^{(j)})\rangle)$, although the analysis will proceed symmetrically if the $S$ gate is applied to compute $\text{Im}(\langle\psi(\theta^{(r)})|\hat{H}|\psi(\theta^{(j)})\rangle)$.

We then apply the $H$ gate again to our quantum system, putting it in the state:

$$\boldsymbol{\Psi} = \frac{1}{2}(|\psi(\theta^{(j)})\rangle|0\rangle + |\psi(\theta^{(j)})\rangle|1\rangle + |\psi(\theta^{(r)})\rangle|0\rangle - |\psi(\theta^{(r)})\rangle|1\rangle) \tag{15}$$

We then compute the expectation value of our Hamiltonian, measuring the effects of our ancilla:

$$\langle\boldsymbol{\Psi}|\hat{H}\otimes Z|\boldsymbol{\Psi}\rangle = \frac{1}{4}(\langle 0|\langle\psi(\theta^{(j)})| + \langle 1|\langle\psi(\theta^{(j)})| + \langle 0|\langle\psi(\theta^{(r)})| - \langle 1|\langle\psi(\theta^{(r)})|)\hat{H}\otimes Z(|\psi(\theta^{(j)})\rangle|0\rangle \tag{16}$$

$$+ |\psi(\theta^{(j)})\rangle|1\rangle + |\psi(\theta^{(r)})\rangle|0\rangle - |\psi(\theta^{(r)})\rangle|1\rangle) \tag{17}$$

$$= \frac{1}{4}(\langle\psi(\theta^{(j)})|\hat{H}|\psi(\theta^{(j)})\rangle + \langle\psi(\theta^{(r)})|\hat{H}|\psi(\theta^{(j)})\rangle - \langle\psi(\theta^{(j)})|\hat{H}|\psi(\theta^{(j)})\rangle \tag{18}$$

$$+ \langle\psi(\theta^{(r)})|\hat{H}|\psi(\theta^{(j)})\rangle + \langle\psi(\theta^{(j)})|\hat{H}|\psi(\theta^{(r)})\rangle + \langle\psi(\theta^{(r)})|\hat{H}|\psi(\theta^{(r)})\rangle \tag{19}$$

$$+ \langle\psi(\theta^{(j)})|\hat{H}|\psi(\theta^{(r)})\rangle - \langle\psi(\theta^{(r)})|\hat{H}|\psi(\theta^{(r)})\rangle) \tag{20}$$

$$= \frac{1}{4}(2\langle\psi(\theta^{(r)})|\hat{H}|\psi(\theta^{(j)})\rangle + 2\langle\psi(\theta^{(j)})|\hat{H}|\psi(\theta^{(r)})\rangle) \tag{21}$$

$$= \frac{1}{4}(4\text{Re}(\langle\psi(\theta^{(r)})|\hat{H}|\psi(\theta^{(j)})\rangle)) \tag{22}$$

$$= \text{Re}(\langle\psi(\theta^{(r)})|\hat{H}|\psi(\theta^{(j)})\rangle) \tag{23}$$

as desired.

## C. Classical finite differences convergence analysis

Following recent theory on nonconvex optimization on manifolds [53], we can restate two main assumptions regarding the convergence of the generic Riemann descent algorithm:

**Assumption 1 (Assumption O.2 in [29])** *(Lower bound). There exists $f^\star > -\infty$ such that $f(x) \geq f^\star$, $\forall x \in \mathcal{M}$.*

**Assumption 2 (Assumption O.3 in [29])** *(Sufficient decrease). There exist scalars $\xi$, $\xi' > 0$ such that, $\forall k \geq 0$,*

$$f(x_k) - f(x_{k+1}) \geq \min\{\xi \cdot \|\nabla^R f(x_k)\|_2, \ \xi'\} \cdot \|\nabla^R f(x_k)\|_2. \tag{24}$$

We make another assumption based on a design decision made in [29], where the authors claim omitting the projection step mimics modulating the learning rate, improving stability:

**Assumption 3** (*Projection*). *For all $k \geq 0, \forall x \in \mathcal{M} = \mathbb{R}^n$,*

$$\nabla^R f(x_k) := \nabla f(x_k). \tag{25}$$

Based on Assumption 3, the sufficient decrease of Assumption 2 may be reformulated as:

$$f(x_k) - f(x_{k+1}) \geq \min\{\xi \cdot \|\nabla f(x_k)\|_2, \ \xi'\} \cdot \|\nabla f(x_k)\|_2. \tag{26}$$

For our derivative-free approach, we will define the gradient of function $f$ as $\nabla f(x) := \nabla^x f(x) + \nabla^\sigma f(x)$, an additively separable gradient that decomposes into a term that corresponds to the analytical gradient $\nabla^x f(x)$, and a term that corresponds to the error $\nabla^\sigma f(x)$. In order to find the number of iterations required to reach convergence, we will consider the case where the algorithm converges with a finite differences gradient estimation (i.e. when the gradient is error-prone, represented by $\nabla f(x)$), and use Theorem 3 of [53] which we state in Theorem 5 of this paper.

**Theorem 5 (Thm 3 in [53])** *Under Assumption 1 and Assumption 2, the generic Riemannian descent algorithm returns an error-prone $x \in \mathcal{M}$ satisfying $f(x) \leq f(x_0)$ and $\|\nabla f(x)\|_2 \leq \delta$ in*

$$\left\lceil \frac{f(x_0) - f^\star}{\xi} \cdot \frac{1}{\delta^2} \right\rceil \tag{27}$$

*iterations, provided $\epsilon \leq \frac{\xi'}{\xi}$. If $\rho > \frac{\xi'}{\xi}$, at most $\lceil \frac{f(x_0) - f^\star}{\xi} \cdot \frac{1}{\delta} \rceil$ iterations are required.*

*Proof.* If Algorithm 2 executes $K - 1$ iterations without terminating, then $\|\nabla f(x_k)\|_2 > \delta$ for all $k \in 0, \ldots, K - 1$. Using Assumption 1 and Assumption 2 in a classic telescoping summation argument gives:

$$
\begin{aligned}
f(x_0) - f^\star \geq f(x_0) - f(x_K) &= \sum_{k=0}^{K-1} f(x_k) - f(x_{k+1}) \\
&\overset{A2}{\geq} \sum_{k=0}^{K-1} \min\{\xi \cdot \|\nabla f(x_k)\|_2, \ \xi'\} \cdot \|\nabla f(x_k)\|_2 \\
&\overset{\|\nabla f(x_k)\|_2 > \delta}{>} \sum_{k=0}^{K-1} \min\{\xi \cdot \delta, \ \xi'\} \cdot \delta \\
&= K \min\{\xi \cdot \delta, \xi'\} \cdot \delta.
\end{aligned}
$$

By contradiction, when $f(x_0) - f^\star \leq K \min\{\xi \cdot \delta, \xi'\} \cdot \delta$, the algorithm will have terminated if $K \geq \frac{f(x_0) - f^*}{\min\{\xi\delta, \xi'\}\delta}$.

The proofs for this section aim towards finding the conditions that satisfy Assumptions 1 and 2, and determine a convergence rate for an $i$-th agent using Theorem 5, which we finally state in Theorem 1. We resume by stating the convergence proofs of [29] that can be directly applied to our problem. Theorem L.1 proves that the PCA solution is the Unique strict-Nash Equilibrium as is stated in the following.

**Theorem 6** (*PCA Solution is the Unique strict-Nash Equilibrium*). *Assume that the top-k eigenvalues of $X^\top X$ are positive and distinct. Then the top-k eigenvectors form the unique strict-Nash equilibrium of the proposed game in Equation (4).*

The theory from Section N of [29] can be directly applied to our problem, as the results presented there mainly depend on the utility function and are algorithm-independent. The main result from the mentioned section is contained in Theorem N.2, where a bound on the angular deviation of any maximizer of a child's utility given any deviation direction for the child or its parents is derived. Here we restate the theorem.

**Theorem 7** *Assume it is given that $|\phi_j| \leq \frac{c_i g_i}{(i-1)\Lambda_{11}} \leq \sqrt{\frac{1}{2}}$ for all $j < i$ with $0 \leq c_i \leq \frac{1}{16}$. Then:*

$$|\phi_i^*| = |\arg\max_{\phi_i}\{u_i(\hat{v}_i(\phi_i, \Delta_i), \hat{v}_{j<i})\}| \leq 8c_i. \tag{28}$$

We can summarize the redefinition of $u_i(\hat{v}_i, v_{j<i})$ included in Section N of [29], which the authors use to determine initialization conditions for $\phi_i$. Lemma N.1 defines $\hat{v}_i = \cos(\phi_i)v_i + \sin(\phi_i)\Delta_i$ in order to restate the utility function using

$$u_i(\hat{v}_i, v_{j<i}) = u_i(v_i, v_{j<i}) - \sin^2(\phi_i)\left(\Lambda_{ii} - \sum_{l>i} z_l \Lambda_{ll}\right). \tag{29}$$

Lemma N.3 further redefines $u_i(\hat{v}_i, v_{j<i})$ as

$$u_i(\hat{v}_i, v_{j<i}) = A(\phi_j, \Delta_j, \Delta_i)\sin^2(\phi_i) - B(\phi_j, \Delta_j, \Delta_i)\frac{\sin(2\phi_i)}{2} + C(\phi_j, \Delta_j, \Delta_i), \tag{30}$$

and Lemma N.4 simplifies the previous expression into

$$u_i(\hat{v}_i, v_{j<i}) = \frac{1}{2}\left[\sqrt{A^2 + B^2}\cos(2\phi_i + \gamma) + A + 2C\right], \tag{31}$$

where $\phi = \tan^{-1}\left(\frac{B}{A}\right)$.

Lemma O.7 also remains unchanged, and specifies an upper bound to the ratio of generalized inner products in Lemma 8 below. By using Lemma O.7 we may also state the Lipschitz bound of the gradient estimate similarly to Lemma O.8 in Lemma 9 below.

**Lemma 8** *Let $|\phi_j| \leq \epsilon < 1$ for all $j < i$. Then, the ratio of generalized inner products is bounded as:*

$$\frac{\langle \hat{v}_i, \Lambda \hat{v}_j \rangle}{\langle \hat{v}_j, \Lambda \hat{v}_j \rangle} \leq \frac{1 + (1 + \kappa_j)\epsilon}{\sqrt{1 - \epsilon^2}}. \tag{32}$$

**Lemma 9** (*Lipschitz Bound*). *Let $|\phi_j| \leq \epsilon < 1$ for all $j < i$. Then, the norm of the approximate ambient gradient of $u_i$ is bounded as:*

$$\|\tilde{\nabla}_{\hat{v}_i} u_i(\hat{v}_i, \hat{v}_{j<i})\|_2 \leq 2\Lambda_{11}\left(1 + (i-1)\frac{1 + (1 + \kappa_{i-1})\epsilon}{\sqrt{1 - \epsilon^2}}\right) + \sigma\left(\|\text{diag}(M)\|_2 + (i-1)\frac{\Lambda_{11}\kappa_{i-1}}{1 - \epsilon^2}\right). \tag{33}$$

*Proof.* Starting with the gradient (eq. 6), we find:

$$\|\tilde{\nabla}_{\hat{v}_i} u_i(\hat{v}_i, \hat{v}_{j<i})\|_2 = \left\|2M\left(\hat{v}_i - \sum_{j<i}\frac{\hat{v}_i^\top M \hat{v}_j}{\hat{v}_j^\top M \hat{v}_j}\hat{v}_j\right) + \sigma\left(\text{diag}(M) - \sum_{j<i}\frac{(M\hat{v}_j)^{\circ 2}}{\hat{v}_j^\top M \hat{v}_j}\right)\right\|_2$$

$$\leq 2\left\|M\left(\hat{v}_i - \sum_{j<i}\frac{\hat{v}_i^\top M \hat{v}_j}{\hat{v}_j^\top M \hat{v}_j}\hat{v}_j\right)\right\|_2 + \sigma\left\|\text{diag}(M) - \sum_{j<i}\frac{(M\hat{v}_j)^{\circ 2}}{\hat{v}_j^\top M \hat{v}_j}\right\|_2 \tag{34}$$

$$\overset{L8}{\leq} 2\Lambda_{11}\left(1 + \sum_{j<i}\frac{1 + (1 + \kappa_j)\epsilon}{\sqrt{1 - \epsilon^2}}\right) + \sigma\left\|\text{diag}(M) - \sum_{j<i}\frac{(M\hat{v}_j)^{\circ 2}}{\hat{v}_j^\top M \hat{v}_j}\right\|_2$$

$$\leq 2\Lambda_{11}\left(1+(i-1)\frac{1+(1+\kappa_{i-1})\epsilon}{\sqrt{1-\epsilon^2}}\right) + \sigma\left\|\text{diag}(M) - \sum_{j<i}\frac{(M\hat{v}_j)^{\circ 2}}{\hat{v}_j^\top M\hat{v}_j}\right\|_2$$

$$\leq 2\Lambda_{11}\left(1+(i-1)\frac{1+(1+\kappa_{i-1})\epsilon}{\sqrt{1-\epsilon^2}}\right) + \sigma\left(\|\text{diag}(M)\|_2 + \sum_{j<i}\left\|\frac{(M\hat{v}_j)^{\circ 2}}{\hat{v}_j^\top M\hat{v}_j}\right\|_2\right)$$

$$\leq 2\Lambda_{11}\left(1+(i-1)\frac{1+(1+\kappa_{i-1})\epsilon}{\sqrt{1-\epsilon^2}}\right) + \sigma\left(\|\text{diag}(M)\|_2 + \sum_{j<i}\|M\hat{v}_j\|_2^2 \cdot \left|\frac{1}{\hat{v}_j^\top M\hat{v}_j}\right|\right)$$

$$\leq 2\Lambda_{11}\left(1+(i-1)\frac{1+(1+\kappa_{i-1})\epsilon}{\sqrt{1-\epsilon^2}}\right) + \sigma\left(\|\text{diag}(M)\|_2 + \sum_{j<i}\frac{\Lambda_{11}^2}{\langle\cos(\phi_j)v_j+\sin(\phi_j)\Delta_j,\Lambda(\cos(\phi_j)v_j+\sin(\phi_j)\Delta_j)\rangle}\right)$$

$$= 2\Lambda_{11}\left(1+(i-1)\frac{1+(1+\kappa_{i-1})\epsilon}{\sqrt{1-\epsilon^2}}\right) + \sigma\left(\|\text{diag}(M)\|_2 + \sum_{j<i}\frac{\Lambda_{11}^2}{\cos(\phi_j)^2\Lambda_{jj}+\sin(\phi_j)^2\langle\Delta_j,\Lambda\Delta_j\rangle}\right)$$

$$\leq 2\Lambda_{11}\left(1+(i-1)\frac{1+(1+\kappa_{i-1})\epsilon}{\sqrt{1-\epsilon^2}}\right) + \sigma\left(\|\text{diag}(M)\|_2 + \sum_{j<i}\frac{\Lambda_{11}^2}{\cos(\phi_j)^2\Lambda_{jj}}\right)$$

$$\leq 2\Lambda_{11}\left(1+(i-1)\frac{1+(1+\kappa_{i-1})\epsilon}{\sqrt{1-\epsilon^2}}\right) + \sigma\left(\|\text{diag}(M)\|_2 + \sum_{j<i}\frac{\Lambda_{11}^2}{\Lambda_{jj}(1-\epsilon^2)}\right)$$

$$= 2\Lambda_{11}\left(1+(i-1)\frac{1+(1+\kappa_{i-1})\epsilon}{\sqrt{1-\epsilon^2}}\right) + \sigma\left(\|\text{diag}(M)\|_2 + \sum_{j<i}\frac{\Lambda_{11}\kappa_j}{(1-\epsilon^2)}\right)$$

$$= 2\Lambda_{11}\left(1+(i-1)\frac{1+(1+\kappa_{i-1})\epsilon}{\sqrt{1-\epsilon^2}}\right) + \sigma\left(\|\text{diag}(M)\|_2 + (i-1)\frac{\Lambda_{11}\kappa_{i-1}}{(1-\epsilon^2)}\right). \tag{35}$$

We use this previous result to provide a Lipschitz bound where the error is upper bounded, and we reach an expression dependent on the current agent $i$.

**Lemma 10** (*Lipschitz Bound with Accurate Parents*) . *Assume* $|\phi_j| \leq \epsilon \leq \frac{c_i g_i}{(i-1)\Lambda_{11}} \leq \sqrt{\frac{1}{2}}$ *for all* $j < i$ *with* $0 \leq c_i \leq \frac{1}{16}$. *Then the norm of the ambient gradient of* $u_i$ *is bounded as:*

$$\|\tilde{\nabla}_{\hat{v}_i}u_i(\hat{v}_i,\hat{v}_{j<i})\|_2 \leq 4\left(\Lambda_{11}i+(1+\kappa_{i-1})cg_i\right) + \sigma\left(\|\text{diag}(M)\|_2 + 2(i-1)\Lambda_{11}\kappa_{i-1}\right) \overset{def}{=} L_i(\sigma) \tag{36}$$

*where* $L_i = 4\left[\Lambda_{11}i+(1+\kappa_{i-1})cg_i\right]$ *when* $\sigma = 0$.

*Proof.* Starting with Lemma 9, we find:

$$\|\tilde{\nabla}_{\hat{v}_i}u_i(\hat{v}_i,\hat{v}_{j<i})\|_2 \leq 2\Lambda_{11}\left(1+(i-1)\frac{1+(1+\kappa_{i-1})\epsilon}{\sqrt{1-\epsilon^2}}\right) + \sigma\left(\|\text{diag}(M)\|_2 + (i-1)\frac{\Lambda_{11}\kappa_{i-1}}{(1-\epsilon^2)}\right)$$

$$\leq 2\Lambda_{11}\left(1+2(i-1)(1+(1+\kappa_{i-1})\epsilon)\right) + \sigma\left(\|\text{diag}(M)\|_2 + 2(i-1)\Lambda_{11}\kappa_{i-1}\right)$$

$$\overset{\text{Assumption}}{\leq} 2\Lambda_{11}\left(1+2(i-1)+2\frac{(1+\kappa_{i-1})cg_i}{\Lambda_{11}}\right) + \sigma\left(\|\text{diag}(M)\|_2 + 2(i-1)\Lambda_{11}\kappa_{i-1}\right)$$

$$\leq 4\left[\Lambda_{11}(1+(i-1))+(1+\kappa_{i-1})cg_i\right] + \sigma\left(\|\text{diag}(M)\|_2 + 2(i-1)\Lambda_{11}\kappa_{i-1}\right)$$

$$= 4\left[\Lambda_{11}i+(1+\kappa_{i-1})cg_i\right] + \sigma\left(\|\text{diag}(M)\|_2 + 2(i-1)\Lambda_{11}\kappa_{i-1}\right). \tag{37}$$

**Corollary 11** (*Bound on Utility*) . *Assume* $|\phi_j| \leq \frac{c_i g_i}{(i-1)\Lambda_{11}} \leq \sqrt{\frac{1}{2}}$ *for all* $j < i$ *with* $0 \leq c_i \leq \frac{1}{16}$. *Then, the norm of the absolute value of the utility is bounded as follows:*

$$|u_i(\hat{v}_i,\hat{v}_{j<i})| = |\hat{v}_i^\top\nabla_{\hat{v}_i}| \leq \|\hat{v}_i\|_2 \cdot \|\nabla_{\hat{v}_i}\|_2 = \|\nabla_{\hat{v}_i}\|_2 \approx \|\tilde{\nabla}_{\hat{v}_i}\|_2 \leq L_i(\sigma), \tag{38}$$

*thereby approximately satisfying Assumption 1.*

**Lemma 12** *Assume* $|\phi_j| \le \frac{c_i g_i}{(i-1)\Lambda_{11}} \le \sqrt{\frac{1}{2}}$ *for all* $j < i$ *with* $0 \le c_i \le \frac{1}{16}$. *Then Assumption 2 is approximately satisfied with* $\xi = \xi' = \frac{8}{5} L_i(\sigma)$. *The proof is developed in Lemma O.10 of* [29].

We now shift our focus towards defining an accuracy for $|\phi_i|$. Lemma 13 defines a somewhat general target accuracy which acts as a criteria for reasonable approximate optimization. Lemma 14 provides upper bounds for the difference between one agent's estimated value for $\theta$ and its optimal value for an inaccurate gradient calculation that includes finite differences. These directly correspond to O.11 and O.12 of [29], respectively. The proof continues with Lemma 15 where a number of iterations is given for convergence of an agent's cost function, and ends on Theorem 1 with the finite sample convergence rate for all agents.

**Lemma 13** (*Approximate Optimization is Reasonable Given Accurate Parents*) . *Assume* $|\phi_j| \le \frac{c_i g_i}{(i-1)\Lambda_{11}} \le \sqrt{\frac{1}{2}}$ *for all* $j < i$ *with* $0 \le c_i \le \frac{1}{16}$. *i.e., the parents have been learned accurately. Then for any approximate local maximizer* $(\bar{\phi}_i, \bar{\Delta}_i)$ *of* $u_i(\hat{v}_i(\phi_i, \Delta_i), \hat{v}_{j<i})$, *if the angular deviation* $|\bar{\phi}_i - \phi_i^*| \le \bar{e}$ *where* $\theta_i^*$ *forms the global max:*

$$|\bar{\phi}_i| \le \bar{e} + 8c_i \tag{39}$$

*where* $\bar{\phi}_i$ *denotes the angular distance of the approximate local maximizer to the true eigenvector* $v_i$.

**Lemma 14** *Assume* $\hat{v}_i$ *is within* $\frac{\pi}{4}$ *of its maximizer, i.e.,* $|\phi_i - \phi_i^*| \le \frac{\pi}{4}$. *Also, assume that* $|\phi_{j<i}| \le \frac{c_i g_i}{(i-1)\Lambda_{11}} \le \sqrt{\frac{1}{2}}$ *with* $0 \le c_i \le \frac{1}{16}$. *Then the norm of the Riemannian gradient of* $u_i$ *upper bounds this angular deviation:*

$$|\phi_i - \phi_i^*| \le \frac{\pi}{g_i} \|\nabla_{\hat{v}_i}^R u_i(\hat{v}_i, \hat{v}_{j<i})\|_2 \tag{40}$$

We now use $\nabla f(x) = \nabla f(x)_x + \nabla f(x)_\sigma$ to state the following lemma.

**Lemma 15** *Assume* $\hat{v}_i$ *is initialized within* $\frac{\pi}{4}$ *of its maximizer and its parents are accurate enough, i.e., that* $|\phi_{j<i}| \le \frac{c_i g_i}{(i-1)\Lambda_{11}} \le \sqrt{\frac{1}{2}}$ *with* $0 \le c_i \le \frac{1}{16}$. *Let* $\delta_i = \rho_i - \|\tilde{\nabla}_{\hat{v}_i} f(\hat{v}_i | \hat{v}_{j<i})_\sigma\|_2$ *be the maximum tolerated error desired for* $\hat{v}_i$. *Then finite-differences Riemannian gradient ascent returns:*

$$|\phi_i| \le \frac{\pi}{g_i} \delta_i + 8c_i, \tag{41}$$

*after at most* $\lceil \frac{5L_i(0)}{4L_i(\sigma)} \cdot \frac{1}{\delta_i^2} \rceil$ *iterations.*

*Proof.* The assumptions of Theorem 5 are approximately met by Corollary 11 and Lemma 12 with $\xi = \xi' = \frac{8}{5}$ using Riemannian gradient ascent. Theorem 5 thus ensures that Riemannian gradient ascent returns unit vector $\hat{v}_i$ satisfying $u(\hat{v}_i) \ge u(\hat{v}_i^0)$ and $\|\nabla^R\|_2 \le \delta_i$ in at most:

$$\left\lceil \frac{u(\hat{v}_i^*) - u(\hat{v}_i^0)}{\frac{8}{5}L_i(\sigma)} \cdot \frac{1}{\delta_i^2} \right\rceil \tag{42}$$

iterations (where $\hat{v}_i$ is initialized to $\hat{v}_i^0$). Also, for any $\hat{v}_i$, $u_i(\hat{v}_i^*) - u_i(\hat{v}_i) \le 2L_i(\sigma)$ where $L_i(\sigma)$ bounds the absolute value of the utility $u_i$ (see Corollary 11) and $\hat{v}_i^* = \arg \max u_i(\hat{v}_i)$. Combining this with Lemma 14 gives:

$$|\phi_i - \phi_i^*| \le \frac{\pi}{g_i} \delta_i \tag{43}$$

after at most $\lceil \frac{5L_i(0)}{4L_i(\sigma)} \cdot \frac{1}{\delta_i^2} \rceil$ iterations. We then use Lemma 13 to yield $|\phi_i| \le \frac{\pi}{g_i}\delta_i + 8c_i$ in the said amount of iterations.

# D. Proof of the main $0^{\text{th}}$-order convergence theorem

We state our finite sample convergence theorem on the $0^{\text{th}}$-order EigenGame for all players.

*(Convergence of $0^{\text{th}}$-order EigenGame for all players). Consider the Algorithm 4 with input matrix $M \in \mathbb{R}^{p \times p}$ and learned "parent" eigenvectors $v_{j<i} \in \mathbb{R}^p$ that are accurate enough, i.e., that $|\phi_{j<i}| \leq \frac{c_i g_i}{(i-1)\Lambda_{11}} \leq \sqrt{\frac{1}{2}}$ with $0 \leq c_i \leq \frac{1}{16}$. Let the initialization vector $v_{i,init}$ be within perturbation $\frac{\pi}{4}$ from $v_i^\star$, i.e., $\angle(v_{i,init}, v_i^\star) \leq \frac{\pi}{4}$, for all $i$. Consider perturbation $\sigma \in \mathbb{R}$ for the finite difference approximation, and step size $\alpha$ for the gradient ascent. Then, Algorithm 4 returns an approximate eigenvector $v_i$ with angular error less than $\phi_{tol} > 0$ in:*

$$T = \left\lceil \mathcal{O}\left( \sum_{i=1}^{k} \frac{L_i(0)}{L_i(\sigma)} \left[ \frac{(k-1)!}{\phi_{tol}} \prod_{j=1}^{k} \left( \frac{16\Lambda_{11}}{g_j} \right) \right]^2 \right) \right\rceil \quad \textit{iterations,}$$

*where $L_i(\sigma)$ is the Lipschitz continuity assumption of the $0^{\text{th}}$-order EigenGame based on a finite difference step size $\sigma$ as in $\|\widetilde{\nabla}_{v_i} f(v_i \mid v_{j<i})\|_2 \leq L_i(\sigma)$, $\Lambda$ is the diagonal eigenvalue matrix of $M$ containing eigenvalues $\Lambda_{11} > \Lambda_{22} > \ldots > \Lambda_{kk}$ with $\Lambda_{11}$ being the top eigenvalue, and $g_i = \Lambda_{ii} - \Lambda_{i+1,i+1}$ is the eigengap between the two consecutive eigenvalues of players $i$ and $i+1$.*

*Proof.* Let $|\phi_{j<i}| \leq \frac{c_i g_i}{(i-1)\Lambda_{11}} \leq \sqrt{\frac{1}{2}}$ with $c_k \leq \frac{1}{16}$ for all $j < k$ and let the initialization vector $v_{i,\text{init}}$ be within perturbation $\frac{\pi}{4}$ from $v_i^\star$, i.e., $\angle(v_{i,\text{init}}, v_i^\star) \leq \frac{\pi}{4}$. Lemma 13 defines a bound for the angular error in $\hat{v}_k$ obtained from Riemmanian gradient descent as:

$$|\bar{\phi}_i| \leq \bar{e} + 8c_i \tag{44}$$

where $\bar{e}$ quantifies the convergence error and $8c_i$ the error propagated by the parents. Let half the error of $|\theta_k|$ come from imperfect parents $\hat{v}_{j<k}$ and half come from the convergence error of the $k$-th agent. Also, let each parent be learned accurately enough, and the error from learning $\hat{v}_{i-1}$ be less than the threshold of any of its succesors. Per [29], the error from $\hat{v}_{i-1}$'s parents can be bounded as

$$c_{i-1} \leq \frac{c_i g_i}{16(i-1)\Lambda_{11}}, \tag{45}$$

which recursively leads to a bound on $c_i$ through:

$$c_i \leq \frac{(i-1)! \prod_{j=i+1}^{k} g_j}{(16\Lambda_{11})^{k-i}(k-1)!} c_k, \tag{46}$$

which would satisfy the requirement for accurate parents in $\hat{v}_i$'s error bound. We may then bound the convergence error of the $i$-th agent as:

$$\rho \leq \left[ \frac{g_i g_{i+1}}{2\pi i \Lambda_{11}} \right] c_{i+1}, \tag{47}$$

with at most

$$t_i = \left\lceil \frac{5 L_i(0)}{L_i(\sigma)} \left( \frac{\pi i \Lambda_{11}}{g_i g_{i+1}} \right)^2 \frac{1}{c_{i+1}^2} \right\rceil \tag{48}$$

iterations through Lemma 15. We can then input eq. 46 to obtain

$$t_i = \left\lceil \frac{5L_i(0)}{L_i(\sigma)} \left( \frac{\pi i \Lambda_{11}}{g_i g_{i+1}} \right)^2 \frac{(16\Lambda_{11})^{2(k-i-1)}((k-1)!)^2}{(i!)^2 \prod_{j=i+2}^k g_j^2} \frac{1}{c_k^2} \right\rceil$$

$$= \left\lceil \frac{5\pi^2 L_i(0)}{L_i(\sigma)} \frac{16^{2(k-i)}\Lambda_{11}^{2(k-i)}((k-1)!)^2}{(\prod_{j=i}^k g_j^2)((i-1)!)^2} \frac{1}{(16c_k)^2} \right\rceil$$

$$\leq \left\lceil \frac{5\pi^2 L_i(0)}{L_i(\sigma)} \left[ \frac{(16\Lambda_{11})^{k-1}(k-1)!}{\prod_{j=1}^k g_j} \frac{1}{16c_k} \right]^2 \right\rceil, \quad \Lambda_{11} \geq g_i \quad \forall i$$

$$= \left\lceil \mathcal{O}\left( \frac{L_i(0)}{L_i(\sigma)} \left[ \frac{(16\Lambda_{11})^{k-1}(k-1)!}{\prod_{j=1}^k g_j} \frac{1}{16c_k} \right]^2 \right) \right\rceil \tag{49}$$

for any agent $i$. The total number of iterations to learn $\hat{v}_{j<k}$ is thus

$$T_k = \left\lceil \mathcal{O}\left( \sum_{i=1}^k \frac{L_i(0)}{L_i(\sigma)} \left[ \frac{(16\Lambda_{11})^{k-1}(k-1)!}{\prod_{j=1}^k g_j} \frac{1}{16c_k} \right]^2 \right) \right\rceil. \tag{50}$$

## E. Parameterized convergence analysis

The *generic Riemannian descent* convergence analysis of [29] is analyzed in the context of a parameterized EigenGame in this section, given that $x = v(\theta) = |\psi(\theta)\rangle$ with $x \in \mathcal{M} = \mathbb{R}^n$, and $\theta \in \mathcal{M} = \mathbb{R}^{\ell \times q}$ over $\ell$ layers and $q$ qubits representing the parameters of an ansatz $U(\theta)$ that prepares a state via $v(\theta) = U(\theta)|s\rangle$. We choose to use $v(\theta)$ instead of $|\psi(\theta)\rangle$ for the proofs to facilitate analysis. Particularly, Assumptions 1 and 2 remain unchanged for $\theta$.

The $\theta$ parameters do not require any explicit projection for $|\psi(\theta)\rangle$ to lie within the unit sphere $\mathcal{S}^{n-1}$, and the design decision for Assumption 3 becomes a strict statement. Hence, we restate Assumption 3 in the following.

**Lemma 16** (*Non-projection*). For all $k \geq 0$, $\forall \theta \in \mathcal{M} = \mathbb{R}^{\ell \times q}$,

$$\nabla^R f(\theta_k) = \nabla f(\theta_k). \tag{51}$$

The proofs for the parameterized EigenGame stems from the *generic Riemannian descent* convergence of Theorem 5, with a similar approach as in the $0^{\text{th}}$-order case, with a rate of convergence for a particular agent $i$ in Lemma 22, and a finite sample convergence rate for all agents in Theorem 2. The remainder of the proofs from Appendix C remain unchanged except for those we restate in the rest of this section.

**Lemma 17** (*Parameterized Lipschitz Bound*) . Let $|\phi_j| \leq \epsilon < 1$ for all $j < i$. Assume $\mathbf{J}_{\hat{\theta}_{ikl}} = \frac{\partial v(\hat{\theta}_i)}{\partial \hat{\theta}_{ikl}} = U'(\hat{\theta}_i)|s\rangle \in \mathbb{C}^n$ where $\left\| \mathbf{J}_{\hat{\theta}_{ikl}} \right\| = 1$. Then the norm of the ambient gradient of $u_i$ is bounded as:

$$\|\nabla_{\hat{\theta}_i} u_i(v(\hat{\theta}_i), v(\hat{\theta}_{j<i}))\| \leq 2\sqrt{\ell q}\Lambda_{11}\left( 1 + (i-1)\frac{1 + (1+\kappa_{i-1})\epsilon}{\sqrt{1-\epsilon^2}} \right). \tag{52}$$

*Proof.* We start by defining the gradient of the utility with respect to $\hat{\theta}_i$ and relating it to the gradient with respect to $v(\hat{\theta}_i)$ as follows:

$$\nabla_{\hat{\theta}_i} u_i(v(\hat{\theta}_i), v(\hat{\theta}_{j<i})) = \frac{du_i}{dv}\frac{dv}{d\hat{\theta}_i} = \mathbf{J}_{\hat{\theta}_i}\nabla_{v(\hat{\theta}_i)} u_i(v(\hat{\theta}_i), v(\hat{\theta}_{j<i})), \tag{53}$$

where

$$
\mathbf{J}_{\hat{\theta}_i} = \underbrace{\left[ - \quad \frac{\partial v(\hat{\theta}_i)}{\partial \hat{\theta}_{ikl}} \quad - \right]}_{\ell \times q}.
$$
(54)

We proceed by including the number of layers $\ell$ and qubits $q$ in the new upper bound:

$$
\begin{aligned}
\|\nabla_{\hat{\theta}_i} u_i(v(\hat{\theta}_i), v(\hat{\theta}_{j<i}))\|_2 &= \left\| \mathbf{J}_{\hat{\theta}_i} \nabla_{v(\hat{\theta}_i)} u_i(v(\hat{\theta}_i), v(\hat{\theta}_{j<i})) \right\|_2 \\
&\leq 2 \left\| \mathbf{J}_{\hat{\theta}_i} \right\|_2 \cdot \left\| M \left( v(\hat{\theta}_i) - \sum_{j<i} \frac{v(\hat{\theta}_i)^\dagger M v(\hat{\theta}_j)}{v(\hat{\theta}_j)^\dagger M v(\hat{\theta}_j)} v(\hat{\theta}_j) \right) \right\|_2 \\
&\stackrel{L9}{\leq} 2 \left\| \mathbf{J}_{\hat{\theta}_i} \right\|_2 \cdot \Lambda_{11} \left( 1 + (i-1) \frac{1 + (1 + \kappa_{i-1})\epsilon}{\sqrt{1-\epsilon^2}} \right) \\
&= 2\sqrt{\ell q}\Lambda_{11} \left( 1 + (i-1) \frac{1 + (1 + \kappa_{i-1})\epsilon}{\sqrt{1-\epsilon^2}} \right).
\end{aligned}
$$
(55)

We now upper bound the error and define a Lipschitz continuity bound for agent $i$.

**Lemma 18** (*Parameterized Lipschitz Bound with Accurate Parents*) . *Assume* $|\phi_j| \leq \epsilon \leq \frac{c_i g_i}{(i-1)\Lambda_{11}} \leq \sqrt{\frac{1}{2}}$ *for all* $j < i$ *with* $0 \leq c_i \leq \frac{1}{16}$, *and* $\mathbf{J}_{\hat{\theta}_{ikl}} = \frac{\partial v(\hat{\theta}_i)}{\partial \hat{\theta}_{ikl}} = U'(\hat{\theta}_i)|s\rangle \in \mathbb{C}^n$ *where* $\left\| \mathbf{J}_{\hat{\theta}_{ikl}} \right\| = 1$. *Then the norm of the ambient gradient of* $u_i$ *is bounded as:*

$$
\|\nabla_{\hat{\theta}_i} u_i(v(\hat{\theta}_i), v(\hat{\theta}_{j<i}))\|_2 \leq 4\sqrt{\ell q} \left[ \Lambda_{11} i + (1 + \kappa_{i-1})c g_i \right] = \sqrt{\ell q} L_i \stackrel{def}{=} L_{\theta_i}.
$$
(56)

*Proof.* Starting with Lemma 17 and using the derivation for the first term of Lemma 10 we find:

$$
\begin{aligned}
\|\nabla_{\hat{\theta}_i} u_i(v(\hat{\theta}_i), v(\hat{\theta}_{j<i}))\|_2 &\leq 2\sqrt{\ell q}\Lambda_{11} \left( 1 + (i-1) \frac{1 + (1 + \kappa_{i-1})\epsilon}{\sqrt{1-\epsilon^2}} \right) \\
&\stackrel{L10}{\leq} 4\sqrt{\ell q} \left( \Lambda_{11} i + (1 + \kappa_{i-1})c g_i \right) \\
&\stackrel{C11}{=} \sqrt{\ell q} L_i.
\end{aligned}
$$
(57)

We redefine the notation of Corollary 11 by accounting for the ansatz parameters $\hat{\theta}$.

**Corollary 19** (*Parameterized Bound on Utility*) . *Assume* $|\phi_j| \leq \frac{c_i g_i}{(i-1)\Lambda_{11}} \leq \sqrt{\frac{1}{2}}$ *for all* $j < i$ *with* $0 \leq c_i \leq \frac{1}{16}$. *Then the norm of the absolute value of the utility is bounded as follows:*

$$
|u_i(v(\hat{\theta}_i), v(\hat{\theta}_{j<i}))| = |v(\hat{\theta}_i)^\dagger \nabla_{v(\hat{\theta}_i)}| \leq \|v(\hat{\theta}_i)\|_2 \cdot \|\nabla_{v(\hat{\theta}_i)}\|_2 = \|\nabla_{v(\hat{\theta}_i)}\|_2 \leq L_i,
$$
(58)

**Lemma 20** *Assume* $|\phi_j| \leq \frac{c_i g_i}{(i-1)\Lambda_{11}} \leq \sqrt{\frac{1}{2}}$ *for all* $j < i$ *with* $0 \leq c_i \leq \frac{1}{16}$. *Then Assumption 2 is approximately satisfied with* $\xi = \xi' = \frac{1}{2L_{\theta_i}}$.

*Proof.* Let $\omega = \alpha \nabla_{\hat{\theta}_i} u_i(\hat{\theta}_i)$, $\alpha > 0$. Using the first-order Taylor approximation of $u_i(\hat{\theta}_i + \omega)$:

$$
\begin{aligned}
u_i(\hat{\theta}_i + \omega) &\approx u_i(\hat{\theta}_i) + \nabla_{\hat{\theta}_i} u_i(\hat{\theta}_i)^\dagger \omega \\
&= u_i(\hat{\theta}_i) + \alpha \nabla_{\hat{\theta}_i} u_i(\hat{\theta}_i)^\dagger \nabla_{\hat{\theta}_i} u_i(\hat{\theta}_i) \\
&= u_i(\hat{\theta}_i) + \alpha \|\nabla_{\hat{\theta}_i} u_i(\hat{\theta}_i)\|^2.
\end{aligned}
$$
(59)

We may now lower bound the utility difference as follows. Let $\alpha = \frac{1}{2L_{\theta_i}}$, then:

$$
\begin{aligned}
u_i(\hat{\theta}_i + \omega) - u_i(\hat{\theta}_i) &\approx u_i(\hat{\theta}_i) + \alpha \|\nabla_{\hat{\theta}_i} u_i(\hat{\theta}_i)\|^2 - u_i(\hat{\theta}_i) \\
&= \alpha \|\nabla_{\hat{\theta}_i} u_i(\hat{\theta}_i)\|^2 \\
&\geq \min(\alpha, \alpha \|\nabla_{\hat{\theta}_i} u_i(\hat{\theta}_i)\|) \|\nabla_{\hat{\theta}_i} u_i(\hat{\theta}_i)\| \\
&= \min(\xi, \xi' \|\nabla_{\hat{\theta}_i} u_i(\hat{\theta}_i)\|) \|\nabla_{\hat{\theta}_i} u_i(\hat{\theta}_i)\|
\end{aligned}
\tag{60}
$$

with $\xi = \xi' = \alpha = \frac{1}{2L_{\theta_i}}$.

**Lemma 21** *Assume $|\hat{v}(\theta_i)\rangle$ is within $\frac{\pi}{4}$ of its maximizer, i.e., $|\phi_i - \phi_i^*| \leq \frac{\pi}{4}$. Also, assume that $|\phi_{j<i}| \leq \frac{c_i g_i}{(i-1)\Lambda_{11}} \leq \sqrt{\frac{1}{2}}$ with $0 \leq c_i \leq \frac{1}{16}$. Then the norm of the Riemannian gradient of $u_i$ upper bounds this angular deviation:*

$$
|\phi_i - \phi_i^*| \leq \frac{\pi}{g_i} \|\nabla_{\hat{\theta}_i} u_i(|v(\hat{\theta}_i)\rangle, |v(\hat{\theta}_{j<i})\rangle)\|
\tag{61}
$$

**Lemma 22** *Assume $|v(\hat{\theta}_i)\rangle$ is initialized within $\frac{\pi}{4}$ of its maximizer and its parents are accurate enough, i.e., that $|\phi_{j<i}| \leq \frac{c_i g_i}{(i-1)\Lambda_{11}} \leq \sqrt{\frac{1}{2}}$ with $0 \leq c_i \leq \frac{1}{16}$. Let $\rho_i$ be the maximum tolerated error desired for $\hat{\theta}_i$. Then Riemannian gradient ascent returns:*

$$
|\phi_i| \leq \frac{\pi}{g_i} \rho_i + 8c_i
\tag{62}
$$

*after at most*

$$
\left\lceil \frac{4L_{\theta_i}^2}{\sqrt{\ell q}} \cdot \frac{1}{\rho_i^2} \right\rceil
\tag{63}
$$

*iterations.*

*Proof.* We input the results of Corollary 19 and Lemma 20 into Lemma 5, thus satisfying Assumptions 1 and 2, respectively, to provide an upper bound on the number of iterations required for Riemannian gradient ascent to reach convergence in the following. Given $\xi = \xi' = \alpha = \frac{1}{2L_{\theta_i}}$ from Lemma 20, we reach a set of parameters $\phi$ that satisfy $u(v(\hat{\theta}_i)) \geq u(v(\hat{\theta}_i^0))$ and $\|\nabla_{\hat{\theta}_i}\| \leq \rho_i$ in at most:

$$
\left\lceil \frac{u(v(\hat{\theta}_i^*)) - u(v(\hat{\theta}_i^0))}{1/2L_{\theta_i}} \cdot \frac{1}{\rho_i^2} \right\rceil
\tag{64}
$$

iterations. We may now use Corollary 19 to specify $u(v(\hat{\theta}_i^*)) - u(v(\hat{\theta}_i^0)) \leq 2L_i = 2L_{\theta_i}/\sqrt{\ell q}$ where:

$$
|\phi_i - \phi_i^*| \leq \frac{\pi}{g_i} \rho_i
\tag{65}
$$

in at most:

$$
\left\lceil \frac{4L_{\theta_i}^2}{\sqrt{\ell q}} \cdot \frac{1}{\rho_i^2} \right\rceil
\tag{66}
$$

iterations.

# F. Proof of the main parameterized convergence theorem

(*Convergence of QuantumGame for all players*). *Algorithm 3 achieves finite sample convergence to within $\phi_{tol}$ angular error of the top-k principal components independent of initialization. Let $|\phi_{j<i}| \leq \frac{c_i g_i}{(i-1)\Lambda_{11}} \leq \sqrt{\frac{1}{2}}$ with $0 \leq c_i \leq \frac{1}{16}$. Let each $v(\hat{\theta}_i) = U(\hat{\theta}_i)|s\rangle$, with a sufficiently expressive ansatz $U(\hat{\theta}_i)$ [46, 47] and an initial state $|s\rangle$ such that $\angle(v(\hat{\theta}_i), v_i^\star) \leq \frac{\pi}{4}$. Algorithm 3 returns the eigenvectors with angular error less than $\phi_{tol}$ in:*

$$
T = \left\lceil \mathcal{O}\left( \sum_{i=1}^{k} \frac{L_{\theta_i}^2}{\sqrt{\ell q}} \left[ \frac{(k-1)!}{\phi_{tol}} \prod_{j=1}^{k} \left( \frac{16\Lambda_{11}}{g_j} \right) \right]^2 \right) \right\rceil \quad \text{iterations,}
$$

where $L_{\theta_i}$ is the Lipschitz continuity constant of QuantumGame, $\ell$ is the number of layers of the ansatz and $q$ is the number of qubits.

*Proof.* Let $|\phi_{j<i}| \leq \frac{c_i g_i}{(i-1)\Lambda_{11}} \leq \sqrt{\frac{1}{2}}$ with $c_k \leq \frac{1}{16}$ for all $j < k$ and let the initialization vector $v_{i,\text{init}}$ be within perturbation $\frac{\pi}{4}$ from $v_i^\star$, i.e., $\angle(v_{i,\text{init}}, v_i^\star) \leq \frac{\pi}{4}$. Using the same convergence error bound for the $i$-th agent:

$$\rho \leq \left\lceil \frac{g_i g_{i+1}}{2\pi i \Lambda_{11}} \right\rceil c_{i+1} \tag{67}$$

from Theorem 1, convergence is reached in at most:

$$t_i = \left\lceil \frac{4 L_{\theta_i}{}^2}{\sqrt{\ell q}} \left( \frac{\pi i \Lambda_{11}}{g_i g_{i+1}} \right)^2 \frac{1}{c_{i+1}^2} \right\rceil \tag{68}$$

iterations using Lemma 22. The total numer of iterations required to reach finite sample convergence for all players with eigenvectors $v(\hat{\theta})_{j<k}$ is then:

$$T_k = \left\lceil \mathcal{O} \left( \sum_{i=1}^{k} \frac{L_{\theta_i}{}^2}{\sqrt{\ell q}} \left[ \frac{(16\Lambda_{11})^{k-1}(k-1)!}{\prod_{j=1}^{k} g_j} \frac{1}{16 c_k} \right]^2 \right) \right\rceil . \tag{69}$$

## G. Error accumulation

**Theorem 23** *Assume the angular error $\phi_j \leq \epsilon$ between the true eigenvector of a parent $v_j$ and its estimate $\hat{v}_j$ to satisfy $\epsilon \ll 1$ such that the length of the cord that connects both vectors is $l = |\hat{v}_j - v_j| = 2\sin(\frac{\epsilon}{2})$ for the Euclidean error of the parent to be $\mathcal{O}(\epsilon)$. The Euclidean error of the child's $0^{th}$-order gradient is $\mathcal{O}(\epsilon)$ and is expressed as:*

$$\|b\| = \|\nabla_{\hat{v}_j}^R u(\hat{v}_j|\hat{v}_{<i}) - \nabla_{\hat{v}_j}^R u(\hat{v}_j|v_{j<i})\|$$
$$\leq 2\|M\| \sum_{j<i} (\|v_j w_j^\top\| + \|w_j v_j^\top\| + \|w_j w_j^\top\|)\frac{\Lambda_{11}}{\Lambda_{jj}} + \sigma \sum_{j<i} (2\|Mv_j\|\|Mw_j\| + \|Mw_j\|^2)\frac{1}{\Lambda_{jj}}. \tag{70}$$

*Proof.* Let $b$ be the difference between the Riemannian gradient with approximate parents and that with exact parents. We can specify $\hat{v}_j = v_j + w_j$ where $w_j$ is the vector that represents the misspecification error of $\hat{v}_j$, and proceed by bounding $b$ in the following:

$$b = \nabla_{\hat{v}_i}^R u(\hat{v}_i|\hat{v}_{<i}) - \nabla_{\hat{v}_i}^R u(\hat{v}_i|v_{j<i}) =$$
$$(I - \hat{v}_i\hat{v}_i^\top) \left[ 2M \left( \hat{v}_i - \sum_{j<i} \frac{\hat{v}_i^\top M \hat{v}_j}{\hat{v}_j^\top M \hat{v}_j} \hat{v}_j \right) + \sigma \left( \texttt{diag}(M) - \sum_{j<i} \frac{(M\hat{v}_j)^{\circ 2}}{\hat{v}_j^\top M \hat{v}_j} \right) - \right.$$
$$\left[ 2M \left( \hat{v}_i - \sum_{j<i} \frac{\hat{v}_i^\top M v_j}{v_j^\top M v_j} v_j \right) + \sigma \left( \texttt{diag}(M) - \sum_{j<i} \frac{(M v_j)^{\circ 2}}{v_j^\top M v_j} \right) \right] = \tag{71}$$
$$\left(I - \hat{v}_i\hat{v}_i^\top\right) \left[ 2M \left( \sum_{j<i} \frac{\hat{v}_i^\top M v_j}{v_j^\top M v_j} v_j - \sum_{j<i} \frac{\hat{v}_i^\top M \hat{v}_j}{\hat{v}_j^\top M \hat{v}_j} \hat{v}_j \right) + \sigma \left( \sum_{j<i} \frac{(M v_j)^{\circ 2}}{v_j^\top M v_j} - \sum_{j<i} \frac{(M\hat{v}_j)^{\circ 2}}{\hat{v}_j^\top M \hat{v}_j} \right) \right],$$

where the norm of the difference is

$$\|b\| = \left\|\left(I - \hat{v}_i \hat{v}_i^\top\right)\left(2M\left(\sum_{j<i}\frac{\hat{v}_i^\top M v_j}{v_j^\top M v_j}v_j - \sum_{j<i}\frac{\hat{v}_i^\top M \hat{v}_j}{\hat{v}_j^\top M \hat{v}_j}\hat{v}_j\right) + \sigma\left(\sum_{j<i}\frac{(Mv_j)^{\circ 2}}{v_j^\top M v_j} - \sum_{j<i}\frac{(M\hat{v}_j)^{\circ 2}}{\hat{v}_j^\top M \hat{v}_j}\right)\right)\right\|_2$$

$$\leq \left\|\left(I - \hat{v}_i \hat{v}_i^\top\right)\right\|_2 \cdot \left\|2M\left(\sum_{j<i}\frac{\hat{v}_i^\top M v_j}{v_j^\top M v_j}v_j - \sum_{j<i}\frac{\hat{v}_i^\top M \hat{v}_j}{\hat{v}_j^\top M \hat{v}_j}\hat{v}_j\right) + \sigma\left(\sum_{j<i}\frac{(Mv_j)^{\circ 2}}{v_j^\top M v_j} - \sum_{j<i}\frac{(M\hat{v}_j)^{\circ 2}}{\hat{v}_j^\top M \hat{v}_j}\right)\right\|_2$$

$$\leq \left\|2M\left(\sum_{j<i}\frac{\hat{v}_i^\top M v_j}{v_j^\top M v_j}v_j - \sum_{j<i}\frac{\hat{v}_i^\top M \hat{v}_j}{\hat{v}_j^\top M \hat{v}_j}\hat{v}_j\right) + \sigma\left(\sum_{j<i}\frac{(Mv_j)^{\circ 2}}{v_j^\top M v_j} - \sum_{j<i}\frac{(M\hat{v}_j)^{\circ 2}}{\hat{v}_j^\top M \hat{v}_j}\right)\right\|_2$$

$$= \left\|2M\left(\sum_{j<i}\frac{\hat{v}_i^\top M v_j}{\Lambda_{jj}}v_j - \sum_{j<i}\frac{\hat{v}_i^\top M \hat{v}_j}{\hat{v}_j^\top M \hat{v}_j}\hat{v}_j\right) + \sigma\left(\sum_{j<i}\frac{(Mv_j)^{\circ 2}}{\Lambda_{jj}} - \sum_{j<i}\frac{(M\hat{v}_j)^{\circ 2}}{\hat{v}_j^\top M \hat{v}_j}\right)\right\|_2$$

$$(72)$$

$$\leq \left\|2M\left(\sum_{j<i}\frac{\hat{v}_i^\top M v_j}{\Lambda_{jj}}v_j - \sum_{j<i}\frac{\hat{v}_i^\top M \hat{v}_j}{\Lambda_{jj}}\hat{v}_j\right) + \sigma\left(\sum_{j<i}\frac{(Mv_j)^{\circ 2}}{\Lambda_{jj}} - \sum_{j<i}\frac{(M\hat{v}_j)^{\circ 2}}{\Lambda_{jj}}\right)\right\|$$

$$\leq 2\|M\|\sum_{j<i}\left\|\frac{\hat{v}_i^\top M v_j}{\Lambda_{jj}}v_j - \frac{\hat{v}_i^\top M \hat{v}_j}{\Lambda_{jj}}\hat{v}_j\right\| + \sigma\sum_{j<i}\left\|\frac{(Mv_j)^{\circ 2}}{\Lambda_{jj}} - \frac{(M\hat{v}_j)^{\circ 2}}{\Lambda_{jj}}\right\|_2$$

$$= 2\|M\|_2 \cdot \sum_{j<i}\left\|\frac{(v_j v_j^\top - \hat{v}_j \hat{v}_j^\top)M\hat{v}_i}{\Lambda_{jj}}\right\|_2 + \sigma \cdot \sum_{j<i}\left\|\frac{(Mv_j)^{\circ 2} - (M\hat{v}_j)^{\circ 2}}{\Lambda_{jj}}\right\|_2$$

$$\leq 2\|M\|_2 \cdot \sum_{j<i}\left\|v_j v_j^\top - \hat{v}_j \hat{v}_j^\top\right\|_2 \cdot \frac{\|M\hat{v}_i\|_2}{|\Lambda_{jj}|} + \sigma \cdot \sum_{j<i}\left\|\frac{(Mv_j)^{\circ 2} - (M\hat{v}_j)^{\circ 2}}{\Lambda_{jj}}\right\|_2$$

$$\leq 2\|M\|_2 \cdot \sum_{j<i}\left\|v_j v_j^\top - \hat{v}_j \hat{v}_j^\top\right\|_2 \cdot \frac{\Lambda_{11}}{\Lambda_{jj}} + \sigma \cdot \sum_{j<i}\left\|\frac{(Mv_j)^{\circ 2} - (M\hat{v}_j)^{\circ 2}}{\Lambda_{jj}}\right\|_2$$

$$= 2\|M\|_2 \cdot \sum_{j<i}\left\|v_j v_j^\top - (v_j + w_j)(v_j + w_j)^\top\right\|_2 \cdot \frac{\Lambda_{11}}{\Lambda_{jj}} + \sigma\sum_{j<i}\left\|\frac{(Mv_j)^{\circ 2} - (M(v_j + w_j))^{\circ 2}}{\Lambda_{jj}}\right\|$$

$$= 2\|M\|_2 \cdot \sum_{j<i}\left\|-(v_j w_j^\top + w_j v_j^\top + w_j w_j^\top)\right\|_2 \cdot \frac{\Lambda_{11}}{\Lambda_{jj}} + \sigma \cdot \sum_{j<i}\left\|\frac{-(2(Mv_j)\circ(Mw_j) + (Mw_j)^{\circ 2})}{\Lambda_{jj}}\right\|_2$$

$$\leq 2\|M\|_2 \cdot \sum_{j<i}(\|v_j w_j^\top\|_2 + \|w_j v_j^\top\|_2 + \|w_j w_j^\top\|_2) \cdot \frac{\Lambda_{11}}{\Lambda_{jj}} + \sigma \cdot \sum_{j<i}(2\|Mv_j\|_2 \cdot \|Mw_j\|_2 + \|Mw_j\|_2^2)\frac{1}{\Lambda_{jj}}.$$

$$(73)$$

**Theorem 24** *Assume the angular error $\phi_j \leq \epsilon$ between the true eigenvector of a parent $v_j$ and its estimate $\hat{v}_j$ to satisfy $\epsilon \ll 1$ such that the length of the cord that connects both vectors is $l = |\hat{v}_j - v_j| = 2\sin(\frac{\epsilon}{2})$ for the Euclidean error of the parent to be $\mathcal{O}(\epsilon)$. The Euclidean error of the child's parameterized gradient is $\mathcal{O}(\epsilon\sqrt{\ell q})$, with $\ell$ being the number of layers and $q$ the number of qubits for the parameter space, and is expressed as:*

$$\|b\|_2 = \|\nabla^R_{\hat{\theta}_i}u(v(\hat{\theta}_i)|v(\hat{\theta}_{j<i})) - \nabla^R_{\hat{\theta}_i}u(v(\hat{\theta}_i)|v(\theta_{j<i}))\|_2$$
$$\leq 2\sqrt{\ell q} \cdot \|M\|_2 \cdot \sum_{j<i}\left(\|v(\theta_j)w_j^\top\|_2 + \|w_j v(\theta_j)^\top\|_2 + \|w_j w_j^\top\|_2\right) \cdot \frac{\Lambda_{11}}{\Lambda_{jj}}. \tag{74}$$

*Proof.* Let $b$ be the difference between the Riemannian gradient with approximate parents and that with exact parents. We can specify $v(\hat{\theta}_j) = v(\theta_j) + w_j$ where $w_j$ is the vector that represents the

mis-specification error of $v(\hat{\theta}_j)$, and proceed by bounding $b$ in the following:

$$
\begin{aligned}
b &= \nabla^R_{\hat{\theta}_i} u(v(\hat{\theta}_i)|v(\hat{\theta}_{j<i})) - \nabla^R_{\hat{\theta}_i} u(v(\hat{\theta}_i)|v(\hat{\theta}_{j<i})) \\
&= (I - v(\hat{\theta}_i)v(\hat{\theta}_i)^\top)\left[\nabla_{\hat{\theta}_i} u(v(\hat{\theta}_i)|v(\hat{\theta}_{j<i})) - \nabla_{\hat{\theta}_i} u(v(\hat{\theta}_i)|v(\hat{\theta}_{j<i}))\right] \\
&= (I - v(\hat{\theta}_i)v(\hat{\theta}_i)^\top)\mathbf{J}_v(\hat{\theta}_i)\left[\nabla_{\hat{\theta}_i} u(v(\hat{\theta}_i)|v(\hat{\theta}_{j<i})) - \nabla_{\hat{\theta}_i} u(v(\hat{\theta}_i)|v(\hat{\theta}_{j<i}))\right] \\
&= (I - v(\hat{\theta}_i)v(\hat{\theta}_i)^\top)\mathbf{J}_v(\hat{\theta}_i)\left[2M\left(v(\hat{\theta}_i) - \sum_{j<i}\frac{v(\hat{\theta}_i)^\top M v(\hat{\theta}_j)}{v(\hat{\theta}_j)^\top M v(\hat{\theta}_j)}v(\hat{\theta}_j)\right) - 2M\left(v(\hat{\theta}_i) - \sum_{j<i}\frac{v(\hat{\theta}_i)^\top M v(\theta_j)}{v(\theta_j)^\top M v(\theta_j)}v(\theta_j)\right)\right] \\
&= \left(I - v(\hat{\theta}_i)v(\hat{\theta}_i)^\top\right)\mathbf{J}_v(\hat{\theta}_i)\left[2M\left(\sum_{j<i}\frac{v(\hat{\theta}_i)^\top M v(\theta_j)}{v(\theta_j)^\top M v(\theta_j)}v(\theta_j) - \sum_{j<i}\frac{v(\hat{\theta}_i)^\top M v(\hat{\theta}_j)}{v(\hat{\theta}_j)^\top M v(\hat{\theta}_j)}v(\hat{\theta}_j)\right)\right],
\end{aligned}
\tag{75}
$$

where the norm of the difference is

$$
\begin{aligned}
\|b\| &= \left\|\left(I - v(\hat{\theta}_i)v(\hat{\theta}_i)^\top\right)\mathbf{J}_v(\hat{\theta}_i)\left[2M\left(\sum_{j<i}\frac{v(\hat{\theta}_i)^\top M v(\theta_j)}{v(\theta_j)^\top M v(\theta_j)}v(\theta_j) - \sum_{j<i}\frac{v(\hat{\theta}_i)^\top M v(\hat{\theta}_j)}{v(\hat{\theta}_j)^\top M v(\hat{\theta}_j)}v(\hat{\theta}_j)\right)\right]\right\| \\
&\le \left\|\left(I - v(\hat{\theta}_i)v(\hat{\theta}_i)^\top\right)\right\|\left\|\mathbf{J}_v(\hat{\theta}_i)\right\|\left\|2M\left(\sum_{j<i}\frac{v(\hat{\theta}_i)^\top M v(\theta_j)}{v(\theta_j)^\top M v(\theta_j)}v(\theta_j) - \sum_{j<i}\frac{v(\hat{\theta}_i)^\top M v(\hat{\theta}_j)}{v(\hat{\theta}_j)^\top M v(\hat{\theta}_j)}v(\hat{\theta}_j)\right)\right\| \\
&\le \left\|\mathbf{J}_v(\hat{\theta}_i)\right\|\left\|2M\left(\sum_{j<i}\frac{v(\hat{\theta}_i)^\top M v(\theta_j)}{v(\theta_j)^\top M v(\theta_j)}v(\theta_j) - \sum_{j<i}\frac{v(\hat{\theta}_i)^\top M v(\hat{\theta}_j)}{v(\hat{\theta}_j)^\top M v(\hat{\theta}_j)}v(\hat{\theta}_j)\right)\right\| \\
&= \left\|\mathbf{J}_v(\hat{\theta}_i)\right\|\left\|2M\left(\sum_{j<i}\frac{v(\hat{\theta}_i)^\top M v(\theta_j)}{\Lambda_{jj}}v(\theta_j) - \sum_{j<i}\frac{v(\hat{\theta}_i)^\top M v(\hat{\theta}_j)}{v(\hat{\theta}_j)^\top M v(\hat{\theta}_j)}v(\hat{\theta}_j)\right)\right\| \\
&\le \left\|\mathbf{J}_v(\hat{\theta}_i)\right\|\left\|2M\left(\sum_{j<i}\frac{v(\hat{\theta}_i)^\top M v(\theta_j)}{\Lambda_{jj}}v(\theta_j) - \sum_{j<i}\frac{v(\hat{\theta}_i)^\top M v(\hat{\theta}_j)}{\Lambda_{jj}}v(\hat{\theta}_j)\right)\right\| \\
&\le 2\|M\|\left\|\mathbf{J}_v(\hat{\theta}_i)\right\|\sum_{j<i}\left\|\frac{v(\hat{\theta}_i)^\top M v(\theta_j)}{\Lambda_{jj}}v(\theta_j) - \frac{v(\hat{\theta}_i)^\top M v(\hat{\theta}_j)}{\Lambda_{jj}}v(\hat{\theta}_j)\right\| \\
&= 2\|M\|\left\|\mathbf{J}_v(\hat{\theta}_i)\right\|\sum_{j<i}\left\|\frac{(v(\theta_j)v(\theta_j)^\top - v(\hat{\theta}_j)v(\hat{\theta}_j)^\top)M v(\hat{\theta}_i)}{\Lambda_{jj}}\right\| \\
&\le 2\|M\|\left\|\mathbf{J}_v(\hat{\theta}_i)\right\|\sum_{j<i}\left\|v(\theta_j)v(\theta_j)^\top - v(\hat{\theta}_j)v(\hat{\theta}_j)^\top\right\|\frac{\|M v(\hat{\theta}_i)\|}{\Lambda_{jj}} \\
&\le 2\|M\|\left\|\mathbf{J}_v(\hat{\theta}_i)\right\|\sum_{j<i}\left\|v(\theta_j)v(\theta_j)^\top - v(\hat{\theta}_j)v(\hat{\theta}_j)^\top\right\|\frac{\Lambda_{11}}{\Lambda_{jj}} \\
&= 2\|M\|\left\|\mathbf{J}_v(\hat{\theta}_i)\right\|\sum_{j<i}\left\|v(\theta_j)v(\theta_j)^\top - (v(\theta_j)+w_j)(v(\theta_j)+w_j)^\top\right\|\frac{\Lambda_{11}}{\Lambda_{jj}} \\
&= 2\|M\|\left\|\mathbf{J}_v(\hat{\theta}_i)\right\|\sum_{j<i}\left\|-(v(\theta_j)w_j^\top + w_j v(\theta_j)^\top + w_j w_j^\top)\right\|\frac{\Lambda_{11}}{\Lambda_{jj}} \\
&\le 2\|M\|\left\|\mathbf{J}_v(\hat{\theta}_i)\right\|\sum_{j<i}(\left\|v(\theta_j)w_j^\top\right\| + \left\|w_j v(\theta_j)^\top\right\| + \left\|w_j w_j^\top\right\|)\frac{\Lambda_{11}}{\Lambda_{jj}} \\
&= 2\sqrt{\ell q}\,\|M\|\sum_{j<i}(\left\|v(\theta_j)w_j^\top\right\| + \left\|w_j v(\theta_j)^\top\right\| + \left\|w_j w_j^\top\right\|)\frac{\Lambda_{11}}{\Lambda_{jj}}.
\end{aligned}
\tag{76}
$$

