# OpenReview forum: "Quantum EigenGame for excited state calculation"
_CPAL.cc/2025/Proceedings_Track — CPAL 2025 (Proceedings Track) Poster_

### Official Review · Reviewer_sCpV · 2025-01-08
**Review of submission 59**

**Rating:** 7
**Confidence:** 2

**Review:**

**Strengths**

1. The use of EigenGame to solve problems in quantum computing was clever and to my knowledge novel. It opens up new directions for more robustly computing excited states, which the authors' do a good job motivating the relevance of.

2. The explanation of the EigenGame was clear and could be followed easily.

3. The accuracy of the 0th order EigenGame was demonstrated in Fig. 6.

4. The algorithms presented were generally clear and the presenting of pseudo-code was helpful.

**Weaknesses**

1. The comparison of the authors method with VQD was not particularly strong. In particular, it was not clear to me: 1) why EigenGame might outperform VQD (since my understanding is that they are optimizing with similar regularizations); 2) whether EigenGame would consistently outperform VQD or whether the result was particular to the authors set-up. Adding more discussion on the differences between EigenGame and VQD would help address the former point, and performing more numerical experiments (with a larger search over $\beta$) would help address the latter point.

2. The paper could occasionally become quite technical. I was not able to follow lines 180-194, where "ancilla" bits, $Z$,  $\mathcal{R}(\omega)$, and S gates are discussed but not defined.

3. While CPAL has a very broad coverage of topics, among which optimization is one, I think this work could benefit from providing more focus on the parsimony and learning goal of CPAL. In particular, it was not clear to me when reading this why CPAL was necessarily the best fit. Maybe one route for this is to discuss the kinds of efficiency benefits expected when implementing a parallel EigenGameR?

**Minor points**

1. What is the $^\circ$ in Eq. 6?
2. Fig. 5 has labels referring to QuantumGame but in the main text the method is referred to as quantum EigenGame.

---

### Official Review · Reviewer_WYcb · 2025-01-08
**Quantum EigenGame for excited state calculation**

**Rating:** 6
**Confidence:** 4

**Review:**

In this paper, the authors introduce a zeroth-order optimization approach for EigenGame and provide an analysis of its convergence rate.

Comments:

1. One notable advantage of quantum circuits is the ability to estimate parameters $\theta$ in quantum gates rather than directly estimating exponentially large eigenvectors. However, in Algorithm 4, the target remains the eigenvector, which necessitates exponentially large storage memory. Could the authors clarify whether it is feasible to modify Algorithm 4 to estimate $\theta$ as the target while still employing zeroth-order optimization techniques?

2. Could the authors elaborate on why the gradient computation is valid as described in line 227?

3. Theorems 1 and 2 suggest that the number of iterations scales exponentially with $k$. Could the authors provide an explanation for why this dependence is exponential rather than polynomial?

4. Why does the paper focus on excited states (higher energy states) rather than lower energy states, such as thermal states at low temperatures? This choice appears unconventional in the context of eigenstate computation, especially considering that, for problems like PCA, it is often assumed that the matrix is low-rank. Could the authors clarify the motivation behind this choice and its implications for the proposed method?

5. In the experimental section, the authors employed the Adam optimizer, whereas Algorithm 4 utilizes gradient descent. To maintain consistency with Theorem 4, it would be more appropriate to apply gradient descent in the experiments.

---

### Meta-Review · Area_Chair_R9MT · 2025-02-02

**Recommendation:** Accept (Poster)
**Confidence:** 3

**Metareview:**

This paper introduces a zeroth-order optimization approach for EigenGame, and it analyzes its convergence rate.
The reviewers valued the paper's presentation and motivation. The authors give a justification for why this paper is a good fit for CPAL.

---

### Decision · Program_Chairs · 2025-02-11

Accept (Poster)